# Automated Membership Inference Attacks: Discovering MIA Signal Computations using LLM Agents

## Abstract

Membership inference attacks (MIAs), which enable adversaries to determine whether specific data points were part of a model's training dataset, have emerged as an important framework to understand, assess, and quantify the potential information leakage associated with machine learning systems. Designing effective MIAs is a challenging task that usually requires extensive manual exploration of model behaviors to identify potential vulnerabilities. In this paper, we introduce `AutoMIA`– a novel framework that leverages large language model (LLM) agents to automate the design and implementation of new MIA signal computations. By utilizing LLM agents, we can systematically explore a vast space of potential attack strategies, enabling the discovery of novel strategies. Our experiments demonstrate `AutoMIA` can successfully discover new MIAs that are specifically tailored to user-configured target model and dataset, resulting in improvements of up to 0.18 in absolute AUC over existing MIAs. This work provides the first demonstration that LLM agents can serve as an effective and scalable paradigm for designing and implementing MIAs with SOTA performance, opening up new avenues for future exploration.[1]

## 1 Introduction

Membership inference attacks (MIAs) are an active area of research that aims to determine whether a specific data point was part of the training dataset of machine learning models (Shokri et al., 2017). Over the last decade, MIAs have been extensively studied and emerged as one of the most widely adopted tools for measuring privacy leakage (Hu et al., 2022). More broadly, MIAs can be viewed as a general mechanism for auditing whether a model retains detectable information about particular training examples, making them relevant beyond privacy to copyrighted-content detection, data provenance analysis, and verification of data removal (machine unlearning). Due to its importance, MIAs have been investigated across a broad range of model architectures, ranging from conventional classification models (Carlini et al., 2022) to recent large language models (LLMs) (Mattern et al., 2023), and across different data modalities, such as vision language (Li et al., 2024), audio (Proboszcz et al., 2026), video (Li et al., 2025), and code (Zhang et al., 2024a). Beyond the model endpoints, model intermediate representations have been shown to be vulnerable to MIAs, such as embeddings (Mahloujifar et al., 2021) and tokenizers (Tong et al., 2026). Despite the significant efforts, MIAs remain a challenging task that often requires domain expertise and careful engineering.

This difficulty arises because each MIA setting shows unique challenges. For example, MIAs on LLMs requires methods to handle the sequential nature of the model responses, making previous MIAs developed for classification models less effective (Wu & Cao, 2025). This requires researchers to manually explore model behaviors and understand memorization patterns to design effective attack strategies. Most prior works (Hu et al., 2022) have relied on manual design driven by domain expertise and intuition. This paper investigates the use of LLM agents to automate the design and implementation process that can adapt to any MIA setting without human intervention. By reducing the human effort, our approach can potentially accelerate the development of MIA methods and explore a larger space of attack and auditing strategies at scale.

---

[1]The code is available at `https://anonymous.4open.science/r/automia-7393/`

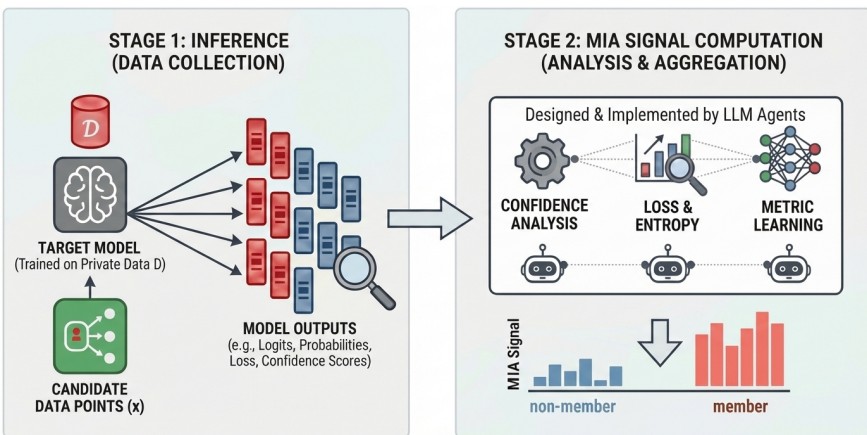

Figure 1: General membership inference attack pipeline. `AutoMIA` employs LLM Agents to design and implement the signal computation strategy.

Despite the diversity of MIA settings, most MIAs follow a common underlying pipeline (illustrated in Fig. 1): (Stage 1: Inference) given a target model and data points, the attacker or auditor conducts queries to the target model and collects the model responses; (Stage 2: MIA Signal Computation) the attacker or auditor applies some aggregation and analysis strategy on the collected data to compute a signal score that can distinguish between members and non-members. Among these stages, the signal computation strategy plays a critical role, as it must amplify the subtle differences between members and non-members. Even small design choices can have an outsized impact, e.g., for MIAs on LLMs, Min-K%++ (Zhang et al., 2025) introduces only a calibration factor on top of Min-K% (Shi et al., 2024), yet achieves significant performance gains. In this paper, we focus on this stage and investigate the use of LLM agents to automate the design and implementation of MIA signal computation strategies.

**Contributions.** We introduce `AutoMIA`, an agentic system for automated MIA design and implementation. Given an MIA setting (e.g., threat model and dataset), `AutoMIA` employs an evolutionary loop in which LLM agents iteratively propose new strategies, implement and evaluate them, and store the results in a shared knowledge base. By learning from previous successes and failures, the system progressively discovers more effective strategies. We envision `AutoMIA` as a general framework for automated privacy evaluation and model auditing, where the goal is to estimate the worst-case privacy leakage or training data retention under user-configured settings. By offloading the design process to LLM agents, `AutoMIA` can explore significantly larger design spaces than manual effort allows, potentially uncovering novel and more effective attacks and audits. We perform experiments for two relatively recent MIA settings on LLMs and Vision Language Models (VLMs). The signal computation strategies designed by `AutoMIA` outperform the baselines in most cases with significant margins up to 0.18 in absolute AUC. Our key contributions are summarized as follows:

- **Proof of concept.** We demonstrate *for the first time* that LLM agents can effectively automate the design and implementation of MIAs, producing attack strategies with state-of-the-art performance. Our work can open new research directions, shifting from manually crafting individual attacks for specific settings to building agentic systems that can adapt to diverse MIA settings and continuously improve over time.

- **System Design.** We introduce `AutoMIA`, an agentic system for automated MIA design and implementation. Unlike general-purpose frameworks such as OpenEvolve which evolve raw code directly, `AutoMIA` evolves high-level attack ideas and designs in natural language for more effective and efficient exploration. We empirically show that `AutoMIA` is more effective compared to OpenEvolve, advancing the MIA performance by up to 0.18 in absolute AUC over human-designed baselines.

- **Novel Attacks and Insights.** The MIA signals found by `AutoMIA` for black-box LLMs and gray-box VLMs are both novel and effective, providing new insights into MIA research for these settings. Our

analysis on the MIA transferability reveals that model memorization patterns can vary significantly across datasets, suggesting that the common practice of proposing a single attack strategy per setting may be insufficient.

## 2 Related Works

**Membership inference attacks.** MIAs have been first introduced by Shokri et al. (2017) to evaluate the privacy risks of machine learning models. In the early stage, MIAs were primarily designed for tabular (Long et al., 2018) and image classification models (Salem et al., 2018; Yeom et al., 2018). With the rise of Generative Adversarial Networks (GANs) (Goodfellow et al., 2014), previous MIAs show limited performance due to the fundamental differences between classification and GAN models, motivating significant efforts to design MIAs for GANs (Hayes et al., 2018; Chen et al., 2020). More recently, diffusion models and large language models (LLMs) have emerged as the state-of-the-art generative models, further motivating new MIAs for these models (Carlini et al., 2021; Mattern et al., 2023; Matsumoto et al., 2023; Pang et al., 2025). The architecture differences between the models have led to the need for designing MIAs tailored for each model type.

Beyond the model architecture, memorization can also differ across different training stages, requiring MIAs to be adapted accordingly. For example, several MIAs have been proposed to target LLM pretraining (Hayes et al., 2026), fine-tuning (Fu et al., 2024b), and alignment (Feng et al., 2025). Additionally, ML models can be deployed in various settings. Each setting has its own constraints, which also lead to new challenges for designing effective MIAs. For example, Choquette-Choo et al. (2021); WU et al. (2024) consider black-box settings, where the adversary can only query the model and observe its predicted classes. In addition to ML models, some MIAs have been used to evaluate the privacy risks of synthetic data (van Breugel et al., 2023; Guépin et al., 2023) and retrieval databases (Liu et al., 2025a; Anderson et al., 2025). All of these MIAs have been developed by experts in the field through analyzing the unique characteristics and memorization patterns of the target model and attack setting, then manually exploring attack strategies. This cycle of "New Setting → Existing MIAs failed → New tailored MIAs" has been observed in the MIA research community for the last decade. In this paper, we explore a new paradigm of automating MIA design and implementation using LLM agents, which can potentially enable the discovery of MIAs across attack settings.

The closest work in this direction is AttackPilot (Wu et al., 2025), which uses LLM Agents to perform inference attacks against machine learning API services. However, AttackPilot aims to reduce engineering effort in implementing MIAs, targeting comparable – rather than superior – performance to existing attacks. In contrast, our goal is to demonstrate the potential of LLM agents in discovering novel attack designs that outperform existing MIAs and take a step towards automating MIA research advancement.

**LLM-guided evolutionary search.** With the increasing capabilities of LLMs over the past few years, Romera-Paredes et al. (2023) was the first to demonstrate the effectiveness of LLM Agents in mathematical discovery. Following this work, AlphaEvolve (Novikov et al., 2025) is a general-purpose agentic coding framework that found a more efficient $4 \times 4$ matrix multiplication algorithm – breaking 56 years of human research. This framework is then used to advance mathematical research (Georgiev et al., 2025), hardware design (Novikov et al., 2025), multi-agent learning algorithms (Li et al., 2026b), computer architecture discovery (Gupta et al., 2026), and compiler optimization (Chen et al., 2026). OpenEvolve (Sharma, 2025) is a community implementation of AlphaEvolve. AlphaEvolve directly evolves bare code, where the LLM agent receives a parent program and some top-performing programs to modify the parent. This general-purpose architecture of AlphaEvolve may not be optimal across all domains. Therefore, several task-specific agentic systems were introduced for neural network architecture search (Liu et al., 2025b), kernel generation (Cao et al., 2026; Andrews & Witteveen, 2025), and query optimization (Handa et al., 2025). To the best of our knowledge, *our work is the first to investigate this direction for MIAs.* `AutoMIA` reasons on the attack high-level ideas and designs in natural language and only then translating the chosen design into code via coding agents.

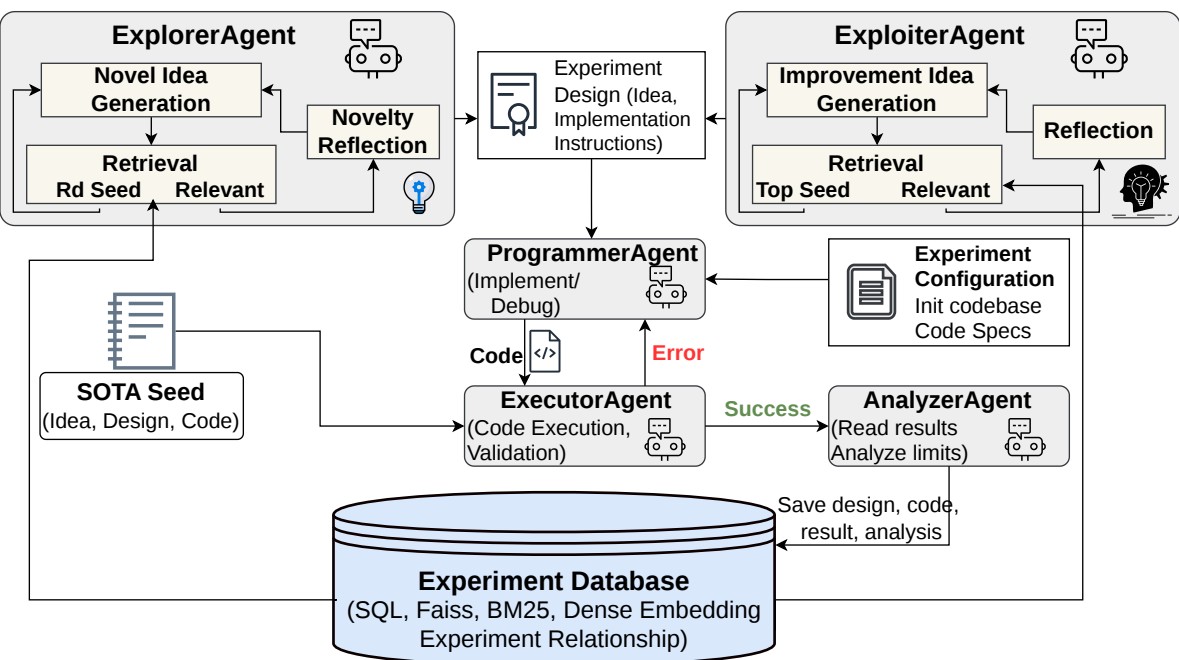

Figure 2: `AutoMIA` architecture. The agents design, implement, and perform experiments, then store attempts into a shared database for future retrieval. This iterative process allows the agents to learn from previous attempts and optimize the MIA designs over time.

## 3 Methodology: `AutoMIA`

**Problem Formulation.** Let $M_\theta$ denote a target model parameterized by $\theta$, trained on a private dataset $D_{train}$. Given a data point $x$, the goal of MIAs is to determine whether $x \in D_{train}$. Following the common MIA pipeline (Fig. 1), let $o$ denote the model output information (e.g., confidence score, logits) of $M_\theta$ on $x$. `AutoMIA` allows users to specify the MIA setting, including the threat model and model access assumptions. Depending on the user configuration, the model output $o$ can be different, ranging from only predicted class for black-box settings to confidence scores or logits in richer-access settings. The attacker needs to design an MIA signal function:

$$f : O \times X \to \mathbb{R},$$

that maps the model output $o$ and the input data point $x$ to a real-valued score $f(o, x)$, the higher the score, the more likely $x$ is a member of $D_{train}$.

Let $D_{MIA_{train}}$ be a dataset used to design the attack, which contains both member and non-member data points. This design process of MIAs can be formulated as the following optimization problem:

$$f* = \arg \max_{f \in \mathcal{F}} \mathcal{J}(f, D_{MIA_{train}}),$$

where $\mathcal{F}$ is the design space of the MIA signal function, and $\mathcal{J}(\cdot)$ is an evaluation metric (e.g., AUC, TPR at low FPR) that measures the attack effectiveness on the design dataset. `AutoMIA` employs LLM agents to traverse $\mathcal{F}$ via an evolutionary search procedure. The final performance is evaluated on a separate test dataset $D_{MIA_{test}}$ to ensure the generalizability.

**System Architecture and Overview.** The main idea of `AutoMIA` is to evolve attack strategies in natural language for more effective and efficient discovery. Fig. 2 shows the architecture of `AutoMIA`. It includes two classes of agents: Design agents (Explorer $\mathcal{A}_{\text{explorer}}$ and Exploiter $\mathcal{A}_{\text{exploiter}}$) and Execution agents (Programmer $\mathcal{A}_{\text{programmer}}$, Executor $\mathcal{A}_{\text{executor}}$, and Analyzer $\mathcal{A}_{\text{analyzer}}$). Design agents are responsible for

generating novel and potential MIA signal designs, while the Execution agents are skilled in translating the designs into executable code, running experiments, and analyzing results.

These agents share a common database $DB$ that stores all experiment attempts. Each attempt is represented as a tuple $s = (id, d, c, r) \in DB$, where $d = (\texttt{idea}, \texttt{design}, \texttt{parent\_id})$ represents the design, $c$ is the code implementation, and $r = (\texttt{status}, \texttt{metrics}, \texttt{analysis})$ is the output after executing the code.

`AutoMIA` first initializes the database $DB$ and sets up the agents with the user-provided configuration $C$. It then runs the seed experiment if available and stores this attempt in the database. After that, the system enters an iterative loop where the Explorer and Exploiter agents alternately generate new designs and optimize existing ones. Each generated design is implemented, executed, and analyzed by the corresponding agents. The designs, results, and insights from each attempt are stored in the database for future reference and retrieval. Over time, the system builds a rich repository of MIA designs, enabling the agents to learn from past attempts and continuously improve the MIA signal designs. The process continues until the pre-defined budget is exhausted. The detailed workflow is summarized in Algo. 3, Sec. A.7.

**User Configuration.** Each MIA setting is defined by a user-provided configuration $C = (\texttt{codebase}, \texttt{spec}, \texttt{params})$. The `codebase` handles model loading, data loading, model inference, and the evaluation function $\mathcal{J}(\cdot)$ that employs the signal computation function $f$. The `spec` describes the specifications of function $f$ (example in Sec. A.6): the structure of the input data (e.g., generated texts, logit tensors) and additional context like data domain. The `params` defines system-level constraints, e.g., the number of attempts, the timeout $T_{max}$ for each attempt, exploration-exploitation schedule. The agents will generate code for function $f$ to fill in the codebase, and execute the code to obtain the performance metrics $\mathcal{J}(f, D_{MIA_{train}})$. Our framework is flexible and supports any MIA setting definable through the configuration. For example, for black-box attacks, the codebase can be designed to only provide labels instead of logits as inputs to the signal computation function $f$.

**Explorer Agent.** The goal of the Explorer $\mathcal{A}_{\text{explorer}}$ is to discover new approaches that have not been tried before. It operates in an iterative novelty-guided loop (Algo. 1, in Sec. A.1.1) and employs three sub-agents: a New Design Generator $LLM_{gen}$, a Novelty Judge $LLM_{judge}$, and a Design Refiner $LLM_{refine}$. At the beginning of each process, the Explorer retrieves random $k$ seed experiments $R \subset DB$ and generates an initial design:

$$d^{(0)} = LLM_{gen}(R, \texttt{spec})$$

The candidate then goes through a refinement loop with a fixed budget. At each iteration $t$, the Explorer retrieves relevant existing designs $E \subset DB$ based on the current design $d^{(t)}$ and evaluates the novelty of the design by comparing it with the retrieved designs:

$$(\texttt{action}, \texttt{suggestion}) = LLM_{judge}(d^{(t)}, E)$$

If $LLM_{judge}$ determines that the design is not novel, it provides suggestions to improve the novelty. The design is then refined based on the feedback from the Novelty Judge:

$$d^{(t+1)} = LLM_{refine}(d^{(t)}, \texttt{suggestion})$$

The process continues until the design is considered as novel or the attempt budget is exhausted. The retrieval employs both dense retrieval (with embeddings of `idea` and `design`) and sparse retrieval (with exact match). If a design is generated by the Explorer, its parent_id is set to `None`.

**Exploiter Agent.** The goal of the Exploiter $\mathcal{A}_{\text{exploiter}}$ is to optimize an existing design by iterative modifications. Let $T = s_1, ..., s_K$ denote the top-K performing experiments in the database $DB$. The Exploiter picks a parent design $d_{parent}$ from $T$ with a probability proportional to their AUC scores:

$$P(d_{parent} = s_i) = \frac{|AUC(s_i) - 0.5|}{\sum_{j=1}^{K} |AUC(s_j) - 0.5|}$$

Given the parent design $d_{parent}$, the Exploiter retrieves its ancestor chain $S_{anc}$, sibling set $S_{sib}$, and semantically relevant designs $S_{rel}$ from the database as references of what has been tried, what has succeeded, and what has failed. The Exploiter is then asked to reason about the references and generate a child design $d_{child}$:

$$d_{child} = LLM_{exploiter}(d_{parent}, S_{anc}, S_{sib}, S_{rel}, \texttt{spec})$$

The tree structure formed by parent-child relationships help to track the design evolution and avoid redundant attempts of sibling designs. The detailed workflow can be found in Algo. 2, in Sec. A.2.1.

**Implementation and Execution Agents.** Once the design is confirmed by either the Explorer or the Exploiter, the Programmer agent $\mathcal{A}_{programmer}$ translates the design $d$ into executable Python code for function $f$:

$$c.\texttt{program} = \mathcal{A}_{programmer}(d, \texttt{spec})$$

The generated code is then executed by the Executor agent $\mathcal{A}_{executor}$, which runs the experiment and collects the results $r$:

$$r = \mathcal{A}_{executor}(c.\texttt{program}, C.\texttt{codebase}, T_{max})$$

If the code execution fails, the error message is sent back to the Programmer agent for debugging and revision. This iterative process continues until the code executes successfully or exceeds the attempt budget to prevent infinite loops. The results of experiments, including performance metrics and any relevant observations, are analyzed by an Analyzer agent $\mathcal{A}_{analyzer}$:

$$r.\texttt{analysis} = \mathcal{A}_{analyzer}(r.\texttt{metrics}, d)$$

The complete attempt, including the design, code, results, and analysis, is stored in the database for future reference.

**Storage and Retrieval.** The database $DB$ is a shared repository that stores all the MIA experiment attempts, including the designs, implementation details, empirical results, and other relevant information. The database is continuously updated with new experiments and serves as a knowledge base for the agents to retrieve information and learn from previous failures and successes. This database supports both dense retrieval (via multi-view embeddings) and sparse retrieval (via keyword matching), allowing the agents to access relevant information efficiently. The database can be implemented using various technologies, such as relational databases, document stores, or vector databases, depending on the specific requirements of the system and the scale of the data.

## 4 Experiments & Results

### 4.1 Overall Evaluation

**Experiment Setup.** We perform experiments on two recent MIA settings including black-box LLMs and gray-box VLMs, which remain underexplored with potential for discovering new MIAs. For each setting, we compare to the existing SOTA human-designed MIAs and a general algorithm search framework - OpenEvolve (Sharma, 2025), which is a community implementation of the original AlphaEvolve (Novikov et al., 2025). We employ Qwen-3-80B-Instruct (Yang et al., 2025) as the backbone LLM for both `AutoMIA` and OpenEvolve. For each setting, we run `AutoMIA` and OpenEvolve for each dataset and model on the training set and *report the performance on the test set*. For fair comparison, all methods employ the same inference stage as the human-design MIAs. Although the framework is easily scalable by parallelization, we run the experiments sequentially. We limit the search time for each setting and model to two hours, with a budget of 100 MIA designs. Each MIA design is timed out after 5 minutes. The details can be found in Sec. B.1.

| Method | ArXiv | | | Github | | | Pubmed | | |
|---|---|---|---|---|---|---|---|---|---|
| | AUC | TPR@ | | AUC | TPR@ | | AUC | TPR@ | |
| | Score | 1%FPR | 5%FPR | Score | 1%FPR | 5%FPR | Score | 1%FPR | 5%FPR |
| | Target model: **Pythia 1.4B** | | | | | | | | |
| Hallinan et al. (2025) | 0.547 | 0.060 | 0.104 | 0.664 | 0.022 | 0.209 | 0.689 | 0.053 | 0.156 |
| OpenEvolve | 0.593 | 0.036 | 0.096 | 0.719 | 0.052 | 0.224 | 0.697 | 0.016 | 0.243 |
| `AutoMIA` | **0.687** | **0.108** | **0.269** | **0.750** | **0.134** | **0.351** | **0.726** | **0.103** | **0.255** |
| | Target model: **OPT 7B** | | | | | | | | |
| Hallinan et al. (2025) | 0.542 | 0.020 | 0.068 | 0.620 | 0.112 | 0.157 | 0.676 | 0.012 | 0.189 |
| OpenEvolve | 0.597 | 0.040 | 0.100 | 0.609 | 0.104 | 0.142 | 0.660 | 0.033 | 0.214 |
| `AutoMIA` | **0.703** | **0.100** | **0.285** | **0.653** | **0.142** | **0.216** | **0.707** | **0.086** | **0.276** |

Table 1: Membership inference attacks on black-box LLMs. The best results are highlighted in **bold**. `AutoMIA` outperforms the baselines across all datasets and target models.

**Black-box MIAs on Large Language Models.** Following the prior works (Hallinan et al., 2025; Duan et al., 2024), we evaluate the MIAs using the MIMIR benchmark. In this setting, the attacker can only access the final generated text from the target LLM without any auxiliary information or access to the model's internal states (e.g., logits, embeddings, or KV cache). The original human-designed MIA is based on the n-gram overlap between the generated text and the ground-truth text (Sec. B.2.2). Tab. 1 shows that both automated systems can discover better MIAs than the human-designed one in most cases. Meanwhile, `AutoMIA` consistently outperforms OpenEvolve, demonstrating the effectiveness of our system design and search algorithm. The performance gain of `AutoMIA` is significant in some cases, e.g., AUC and TPR at 1% FPR of OPT-7B on ArXiv increase from 0.54 to 0.7 and 0.02 to 0.1, respectively. Depending on the dataset and model, the performance gain varies, potentially due to different risk levels, room for improvement, and complexity of the MIA design space. While the human design is straightforward by using n-gram matching, the proposed MIAs by `AutoMIA` are *creative and non trivial*, e.g., geometric edit-distance (Sec. B.2.3). The best-performing MIAs on ArXiv and Pubmed utilize edit-distance signals, while the best on Github considers rare n-gram signals. This suggests that MIAs should be tailored to the target context, as the memorization behavior of LLMs can vary across scenarios.

| Method | Image Logits | | | Text Logits | | |
|---|---|---|---|---|---|---|
| | AUC | TPR@1%FPR | TPR@5%FPR | AUC | TPR@1%FPR | TPR@5%FPR |
| | Dataset: **DALL-E** | | | | | |
| Blind baseline | AUC 0.796 \| TPR@1%FPR 0.122 \| TPR@5%FPR 0.250 | | | | | |
| Li et al. (2024) | 0.594 | **0.054** | 0.135 | 0.605 | **0.007** | 0.142 |
| OpenEvolve | 0.612 | 0.027 | 0.135 | 0.575 | **0.007** | 0.097 |
| `AutoMIA` | **0.752** | **0.054** | **0.304** | **0.705** | **0.007** | **0.209** |
| | Dataset: **Flickr** | | | | | |
| Blind baseline | AUC 0.944 \| TPR@1%FPR 0.679 \| TPR@5%FPR 0.780 | | | | | |
| Li et al. (2024) | 0.590 | 0.031 | 0.132 | **0.736** | 0.075 | 0.164 |
| OpenEvolve | 0.725 | 0.013 | 0.082 | 0.636 | 0.057 | 0.208 |
| `AutoMIA` | **0.770** | **0.044** | **0.189** | 0.719 | **0.145** | **0.277** |

Table 2: Membership inference attacks on gray-box VLMs. The best results are highlighted in **bold**. `AutoMIA` can effectively attack both image and text logits. The blind baseline (Das et al., 2025) is a trained classifier using Dinov2 features (Oquab et al., 2024), which quantifies the distribution shift between the member and non-member samples.

**Gray-box MIAs on Vision Language Models.** We reuse the setting from the prior work (Li et al., 2024), where the attacker has access to the target model's output logits but not its weights, gradients, or intermediate representations. The original work calculate Renyi entropy as the MIA score (Sec. B.3.3). Tab. 2 shows that `AutoMIA` can discover better MIAs than the human design and OpenEvolve in most cases. `AutoMIA` boost the AUC of the image logits from 0.59 by the baseline to 0.75 in both datasets. This

demonstrates the potential of `AutoMIA` in revealing new privacy vulnerabilities that were not revealed by the existing manually designed MIAs. Some of the designs proposed by `AutoMIA` are novel. For example, the rank-stability signal (Sec. B.3.4) adds noise to the logits and measures whether the token prediction rank remains stable. *Although AutoMIA enhances the performance on these datasets, this benchmark has been shown to exhibit a significant distribution shift between the member and non-member sets (Das et al., 2025). Thus, the MIA performance of all methods including the human-designed MIAs can be the joint effect of both the distribution shift and the memorization signal. We leave the exploration of more realistic benchmarks for future work.*

**Black-box MIAs on Large Reasoning Models.** We perform an evaluation on a recent MIA setting on Large Reasoning Models (LRMs) (Hu et al., 2026). In this setting, the target model exposes its reasoning trace through the API. The attacker only observes the reasoning trace and final response, without access to logits, model parameters, or intermediate representations, and must determine membership based solely on this information. Tab. 3 provides the results of this MIA setting. Additional to the SOTA human-designed MIA (Hu et al., 2026), we include a few baselines that utilize the reasoning trace length, the compression ration, and the likelihood by a language model. The results show that `AutoMIA` consistently achieves the best performance on both datasets, improving AUC from 0.777 to 0.827 on ArXiv and from 0.799 to 0.843 on Book. It also substantially increases TPR at low FPR, demonstrating stronger membership inference performance than all baselines.

| Method | ArXiv | | | Book | | |
|---|---|---|---|---|---|---|
| | AUC | TPR@1%FPR | TPR@5%FPR | AUC | TPR@1%FPR | TPR@5%FPR |
| Reasoning Len | 0.507 | 0.005 | 0.040 | 0.542 | 0.000 | 0.054 |
| Gzip Ratio | 0.553 | 0.016 | 0.069 | 0.540 | 0.014 | 0.090 |
| GPT2 NLL | 0.585 | 0.024 | 0.079 | 0.693 | 0.107 | 0.239 |
| Hu et al. (2026) | 0.777 | 0.077 | 0.311 | 0.799 | 0.082 | 0.295 |
| `AutoMIA` | **0.827** | **0.129** | **0.404** | **0.843** | **0.185** | **0.433** |

Table 3: Performance comparison on MIAs for Large Reasoning Models on the ArXiv and Book datasets.

## 4.2 Ablation Study

We analyze the impact of several components of `AutoMIA`. Tab. 4 shows the results on the black-box MIAs for OPT-7B on ArXiv. First, we study of the representation of MIA experiments (denoted as *w/o Idea & Thoughts*). We compare our design (high-level idea, design, code, and result summary) with a simple alternative by AlphaEvolve (code and score). Without the natural language description, the agents may not effectively reason about previous failures and successes to propose better designs, leading to a large performance drop (AUC drops from 0.7 to 0.59).

| System | AUC | TPR@ 5% FPR |
|---|---|---|
| `AutoMIA` (full) | 0.703 | 0.285 |
| w/o Idea & Thoughts | 0.594 | 0.120 |
| w/o Exploiter | 0.637 | 0.072 |
| w/o Explorer | 0.674 | 0.177 |

Table 4: Ablation study of `AutoMIA`.

Furthermore, we analyze the impact of the design agents by removing each of them (denoted as *w/o Explorer* and *w/o Exploiter*). The performance drops more significantly without the exploiter agent, which indicates the importance of attack refinement. The explorer agent also contributes to the performance. Combining both exploration and exploitation yields the best performance as they complement each other during the search process.

### 4.3 Findings and Analyses

**Transferability.** We discover MIAs on ArXiv and evaluate their transferability to other datasets. Fig. 3 shows that the best performing MIA on ArXiv can transfer well to some datasets (e.g., Mathematics, Pubmed, Hackernews), but not all (e.g., Github). This suggests that LLM memorization behavior varies across data characteristics and domains, highlighting the necessity of dataset-specific MIA discovery.

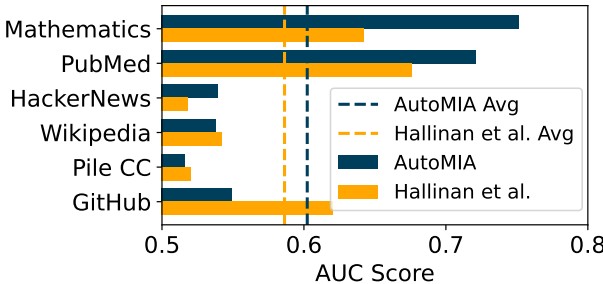

Figure 3: Transferability of the MIA discovered on ArXiv to other datasets. Some datasets have good transferability, while others do not.

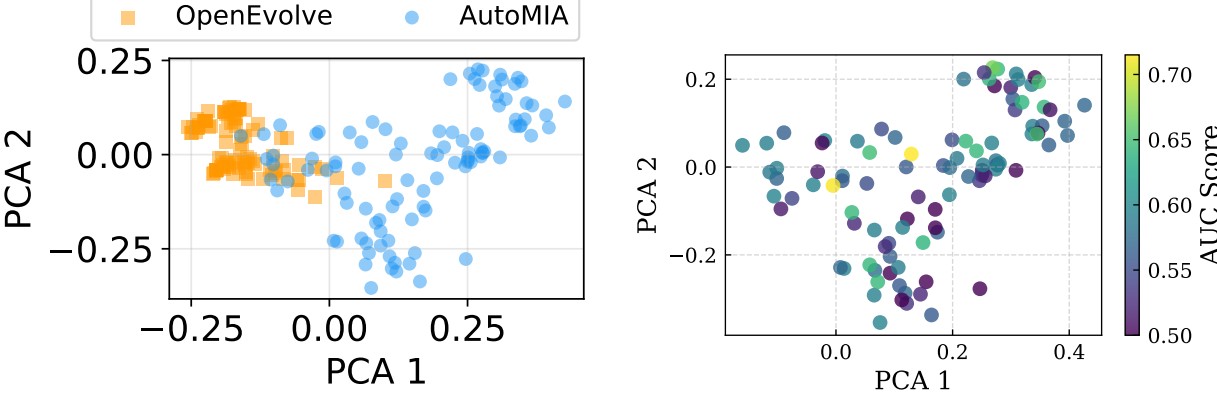

(a) Discovered MIAs on the shared PCA space. Each point represents an MIA design. `AutoMIA` explores more broadly than OpenEvolve.

(b) MIA performance on PCA space. Each point represents an MIA design by `AutoMIA`, and the color indicates its performance (AUC). Each high-performing MIA can be surrounded by low-performing MIAs.

Figure 4: PCA analysis of discovered MIA designs.

**Diversity.** We embed the MIAs by extracting the design choices using zero-shot prompting, detailed in Sec. B.4.1. We then visualize the MIAs in the PCA space (Fig. 4a) and calculate the pairwise cosine similarity within each system's discovered set (Fig. 7 in Sec. B.4.1). Both indicates that `AutoMIA` discovers more diverse MIAs than OpenEvolve, which may contribute to the better performance of `AutoMIA`. This advantage can come from the system design of `AutoMIA`, which evolves the algorithms from high-level descriptions for better and more efficient exploration, while OpenEvolve evolves directly from the code.

Fig. 4b illustrates that each high-performing MIA by `AutoMIA` is surrounded by low-performing MIAs in the PCA space. This suggests that the MIA performance landscape is complex and non-smooth. Consequently, discovering high-performing MIAs requires careful design, tuning, and exploration combining with exploitation.

**Evolving from scratch.** We consider `AutoMIA` without the seed MIA (i.e., existing human-designed MIA) for black-box LLMs. Fig. 5a provides the evolution over iterations in both cases. While the seeded `AutoMIA`

can evolve faster in the early iterations as it can leverage the seed MIA, the end performance of both approaches is approximately equivalent (AUC gap $< 1\%$). This suggests that `AutoMIA` can potentially discover effective MIA signals from scratch without relying on existing human knowledge, which is promising for discovering new MIAs in novel and open settings.

**Target Context.** In the overall evaluation, we do not provide the target model and dataset information to the agents, as we want to test `AutoMIA`'s general capability. This study focuses on the Github dataset, which presents unique challenges of code snippets. We compare `AutoMIA` with and without the context (detailed in Sec. B.4.2). Fig. 5b demonstrates that although the top 1 MIAs' performance are not significantly different, `AutoMIA` with the context provides higher-quality MIAs at most iterations. By leveraging the context, the agents can better understand and reason to avoid proposing MIAs that are general for natural language and focus on code characteristics (examples in Sec. B.4.2).

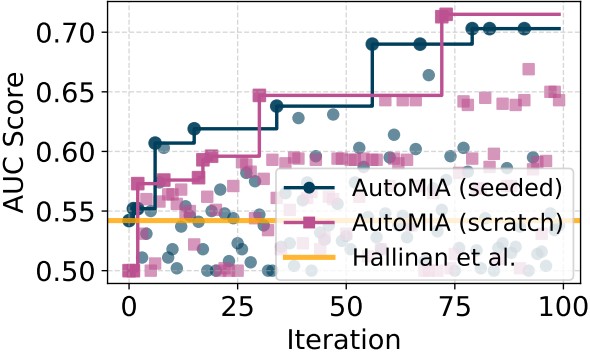 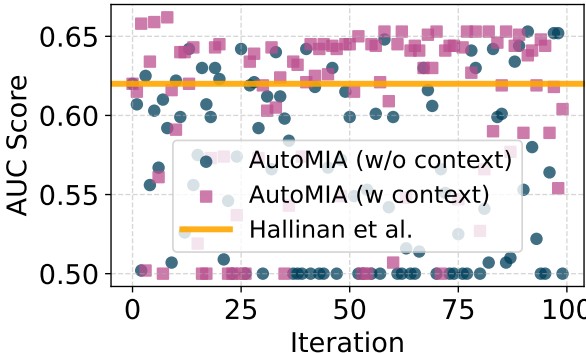

(a) `AutoMIA` with and without the human-designed MIA as seed. The performance is approximately equivalent. Each point represents an MIA design. The lines connect the best-so-far MIA designs over iterations.

(b) `AutoMIA` with and without target context. `AutoMIA` with target context produces higher-quality MIAs at most iterations.

Figure 5: Ablations on seeding and target context.

**LLM Backbone.** We run `AutoMIA` with a strong closed-source LLM backbone. Fig. 6 shows that `Claude Haiku-4.5` found a stronger MIA than `Qwen-3-80B`. This suggests the potential of using stronger LLMs to discover more effective MIAs.

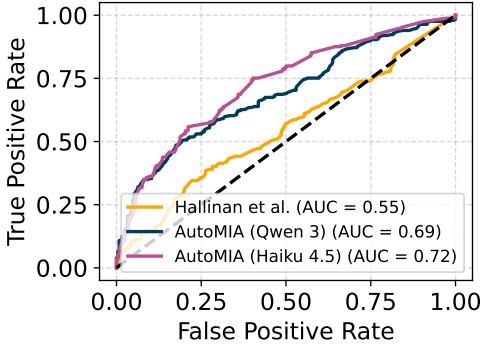

Figure 6: `AutoMIA` with different LLM backbones for Pythia-1.4B on Pubmed. `AutoMIA` with `Claude Haiku-4.5` performs better than `Qwen-3-80B`.

**Exploration-Exploitation Ratio.** We vary the ratio of exploration and exploitation in `AutoMIA`, each ratio setting of `AutoMIA` provides 100 MIA designs for the black-box LLMs on ArXiv. We analyze the

performance of the median and top-performing MIAs among the 100 proposed MIAs. We found that 1/3 of the budget for exploration and 2/3 for exploitation yields the best performance for both median and top-performing attacks among the 100 proposed MIAs, presented in Fig. 9. This suggests that a balanced exploration-exploitation strategy is crucial for discovering effective MIAs, as it allows the system to explore diverse designs while refining promising candidates.

**AutoMIA vs. Supervised MIAs.** Another attack approach is *supervised* MIAs, which train a classifier to predict whether a sample is in the training set. While the main point of `AutoMIA` is to automate the manual efforts in feature crafting and signal design, outputing *unsupervised* MIAs. We do not treat supervised MIAs as a direct baseline for `AutoMIA`, as they operate under a different threat model, and their reliance on learned features limits transferability.

We conduct experiments for the black-box LLM MIA setting, where we train a supervised classifier on the raw n-gram features before aggregation proposed by Hallinan et al. (2025). We then evaluate the transferability across models and benchmarks. Even within the same distribution (i.e., same model and benchmark), the supervised MIA does not necessarily outperform the well-designed unsupervised signal aggregation methods, detailed in Sec. B.4.3. This indicates that given the large number of features and limited training data, supervised classifier may not be able outperform well-crafted aggregation methods. When transferring to other models and benchmarks, the supervised MIA's performance drops significantly, while `AutoMIA` and the human-designed MIAs maintain the performance.

The common assumption in the MIA community is that such unsupervised signals generalize beyond the data they were designed on, letting the human baselines report performance without a held-out test set. As a new approach, rather than take this for granted, we make the design data explicit and enforce a strict train/test split.

## 5 Conclusion

We have introduced `AutoMIA`– an agentic system that effectively automates the design and implementation of MIA signal computations. Through an evolutionary loop, `AutoMIA` iteratively proposes diverse attack strategies, evaluates them, and leverages the results to refine its approach. Our experiments show that `AutoMIA` can discover novel MIA signal computation methods that outperform existing baselines across settings, eliminating the need for manual, setting-specific engineering that has traditionally driven MIA research. To the best of our knowledge, this is the first work to demonstrate the effectiveness of agentic systems to automate the design and implementation of MIAs. Our work represents a significant step towards automating MIA research, opening new research directions in this area – shifting from manual, human-driven attack and auditing design toward automated systems that can adapt to diverse settings and continuously improve over time.

## Limitations

We acknowledge several limitations of our work that future research could address.

**Design Scope.** `AutoMIA` has demonstrated its effectiveness, but it is currently focusing on the signal computation step of the MIA pipeline only. Future work could explore extending `AutoMIA` to end-to-end automated attacks, in which the agentic system can decide on the query strategy, though the computational cost for this step is significantly higher, especially for large models.

**Execution Infrastructure.** `AutoMIA` uses a simple code implementation and execution flow, leaving substantial room for improvement compared to leading coding agentic systems such as Claude Code, Codex, Gemini-CLI, and OpenCode. Additionally, our current execution infrastructure is based on a pre-installed environment with a set of common libraries. This contributes to the failure of executing some implementations. We acknowledge that around 15-20% of the proposed designs could not be implemented and executed suc-

cessfully, which limits the overall performance of `AutoMIA`. Future work could explore more advanced agentic coding frameworks and robust execution infrastructure to further improve the performance of `AutoMIA`.

**Limited Benchmark.** While we evaluated two recent and understudied MIA settings, MIAs are a broad area with many different settings. The proposed framework itself is general and can be applied to other MIA settings. Future work could explore more settings, especially models that are domain-specific (e.g., healthcare and finance) to assess whether LLM agents can advance the state-of-the-art in those domains. While we envision `AutoMIA` for automated MIA red teaming that finds dataset-specific attack strategies, it can be also used to find general attack strategies across domains by replacing the user-configured evaluation function. Additionally, we limit `AutoMIA` at 100 iterations and 2 hours of run time for computational efficiency, the more iterations, the more attack strategies `AutoMIA` can explore, which likely yields better performance. Future work could scale up the system by running the process in a distributed manner.

**Dual-use and broader applicability.** While this paper frames `AutoMIA` as a tool for auditing and red-teaming, the same system could make it easier for adversaries to find stronger attacks against currently deployed models. Additionally, while we examine `AutoMIA` for only MIAs, the key idea of `AutoMIA`'s architecture – reasoning over the design space of attack strategies and iteratively improving them – can be applied to other types of attacks and research problems such as adversarial attacks (Carlini et al., 2025) and machine-generated text detection (Zhang et al., 2024b; Guo et al., 2026; Li et al., 2026a).

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

# Appendix for "Automated Membership Inference Attacks: Discovering MIA Signal Computations using LLM Agents"

# A AutoMIA's Details

## A.1 Explorer

### A.1.1 Novelty-Guided Signal Design Loop

---

**Algorithm 1** Explorer Agent: Novelty-Guided Signal Design Loop

---

**Require:** Database $\mathcal{D}$, iteration budget $B$
**Ensure:** Novel MIA signal design $d$

    *— Generate initial candidate —*
1: $\mathcal{R} \leftarrow$ sample $k$ random experiments from $\mathcal{D}$
2: $d \leftarrow \text{NEWDESIGNLLM}(\mathcal{R})$

    *— Novelty-guided refinement loop —*
3: **for** $i = 1$ **to** $B$ **do**
      // *Retrieve nearest neighbours for novelty check*
4:      $\mathcal{N}_{\text{idea}} \leftarrow \text{SEMANTICNN}(d.\text{idea},\ \text{field} = \text{idea},\ k{=}2)$
5:      $\mathcal{N}_{\text{just}} \leftarrow \text{SEMANTICNN}(d.\text{justification},\ \text{field} = \text{justification},\ k{=}2)$
6:      $\mathcal{N}_{\text{anal}} \leftarrow \text{SEMANTICNN}(d.\text{justification},\ \text{field} = \text{analysis},\ k{=}2)$
7:      $\mathcal{N}_{\text{bm25}} \leftarrow \text{BM25}(d.\text{idea} \oplus d.\text{justification},\ k{=}5)$
8:      $\mathcal{N} \leftarrow \text{DEDUP}(\mathcal{N}_{\text{idea}} \cup \mathcal{N}_{\text{just}} \cup \mathcal{N}_{\text{anal}} \cup \mathcal{N}_{\text{bm25}})$
      // *Judge novelty*
9:      $(action,\ score,\ suggestions) \leftarrow \text{NOVELTYJUDGELLM}(d,\ \mathcal{N})$
10:    **if** $action = \texttt{accept}$ **then**
11:       **break**                                  ▷ Design is sufficiently novel
12:    **else if** $action = \texttt{revise}$ **then**
13:       $d \leftarrow \text{REVISEDESIGNLLM}(d,\ \mathcal{N},\ suggestions)$
14:    **else if** $action = \texttt{redesign}$ **then**
15:       $\mathcal{R} \leftarrow$ sample $k$ random experiments from $\mathcal{D}$
16:       $d \leftarrow \text{NEWDESIGNLLM}(\mathcal{R})$                  ▷ Start from scratch
17:    **end if**
18: **end for**
19: **return** $d$

---

### A.1.2 Explorer Agent Prompt Templates

---

**(Explorer) New-Design Generator**

**System:**

You are an expert researcher designing Membership Inference Attack (MIA) signals. Your goal is to propose a **novel** and **effective** signal that can distinguish member samples from non-member samples using only the available inputs and context. Prefer signals that are robust (e.g., to paraphrasing/noise) and not a trivial rewrite of the example.

**User:**

Design a new MIA signal calculation method according to the spec below.
Provide the following information. Please be concise and only provide key points:
- High-level idea
- Design justification (max 300 words)
- Implementation instructions
`{experiment_context}`

---

**Function specifications:**
{function_description}

**Reference example** (for format/style only; do NOT copy its approach):
{example_python_code}

**Previous attempts** (do NOT copy):
{example_mia_signal_designs}

**Hard constraints** for the eventual code implementation:
- Python only, executable.
- Do NOT include any main/test functions in the code implementation.
- Strictly follow the function specifications and input/output specifications. Do NOT deviate from the function specifications and input/output specifications.
- Easy to read and easy to modify.

**Additional guidance:**
- The example code represents the current state of the art in MIA signal design. You MUST propose a better MIA signal design to outperform the SOTA.
- You can either propose a new design that inspires from the example code and previous attempts, or propose a completely novel design that is completely different from the example code and previous attempts.
- Please refer to the example code for format and input/output specifications.
- Keep the signal computationally reasonable for many samples.
- Make reasonable default choices of hyperparameters.
- Keep implementation instructions brief and focused on the core algorithm.
- The proposed signal MUST be different and significant from the previous attempts and likely to outperform them.

Output a JSON with exactly these fields:
- `idea`: The high-level idea of the MIA signal
- `design_justification`: The design justification of the MIA signal
- `implementation_instruction`: The implementation instruction of the MIA signal

---

**(Explorer) Novelty Judge**

**System:**

You are a strict novelty checker for MIA signal designs. Do NOT accept unless similarity is low and the core mechanism is new. Compare the candidate against prior attempts and decide if it is significantly meaningful different from the prior attempts.

**User:**

Candidate design:
- Idea: {idea}
- Design justification: {design_justification}
- Implementation instructions: {implementation_instruction}

**Nearest prior attempts** (most similar first):
{relevant_mia_signal_designs}

Return a JSON with exactly these fields:
- `action`: one of ['accept', 'revise', 'redesign']. `accept` means the candidate is novel enough and should be implemented and run as a new experiment. `revise` means the candidate is not novel enough but can be revised to be novel. `redesign` means the candidate is too similar to the prior attempts, the general approach is not promising, and the entire proposed approach should be redesigned.

- `reasons`: brief reasoning for the action.
- `novelty_score`: float in $[0, 1]$ where $0$ = identical, $1$ = unexplored.
- `suggestions`: if action is '`revise`', list concrete changes and directions to make it novel. Otherwise, leave this field as an empty string.

---

### (Explorer) Design-Refining Agent

**System:**

You are an expert researcher designing Membership Inference Attack (MIA) signal designs. Revise the provided candidate so it becomes meaningfully more novel and effective than prior attempts. Be concise, keep revisions targeted, and avoid trivial rewrites.

**User:**

Design a new MIA signal calculation method according to the spec below.

**Function specifications:**
`{function_description}`

**Reference example** (for format/style only; do NOT copy its approach):
`{example_python_code}`

**Relevant previous attempts:**
`{relevant_mia_signal_designs}`

**Current candidate:**
`{current_design}`

**Feedback from the nearest neighbor checker:**
`{feedback}`

Provide the following information. Please be concise and only provide key points:
- High-level idea
- Design justification (max 300 words)
- Implementation instructions
`{experiment_context}`

**Hard constraints** for the eventual code implementation:
- Python only, executable.
- Do NOT include any main/test functions in the code implementation.
- Strictly follow the function specifications and input/output specifications. Do NOT deviate from the function specifications and input/output specifications.
- Easy to read and easy to modify.

**Additional guidance:**
- The example code represents the current state of the art in MIA signal design. You MUST propose a better MIA signal design to outperform the SOTA.
- Please refer to the example code for format and input/output specifications.
- You can either propose a new design that inspires from the example code and previous attempts, or propose a completely novel design that is completely different from the example code and previous attempts.
- Keep the signal computationally reasonable for many samples.
- Make reasonable default choices of hyperparameters.
- Keep implementation instructions brief and focused on the core algorithm.
- The proposed signal MUST be different and significant from the previous attempts and likely to outperform them.

Output a JSON with exactly these fields:

- `idea`: The high-level idea of the MIA signal
- `design_justification`: The design justification of the MIA signal
- `implementation_instruction`: The implementation instruction of the MIA signal

## A.2 Exploiter

### A.2.1 Performance-Guided Design Refinement

---

**Algorithm 2** Exploiter Agent: Performance-Guided Design Refinement

---

**Require:** Database $\mathcal{D}$ of past experiments with scores
**Ensure:** Refined MIA signal design $d$
1: Wait until $\mathcal{D}$ contains at least one scored experiment
2: $\mathcal{T} \leftarrow$ top-$k$ experiments from $\mathcal{D}$ ranked by AUC
3: Cluster $\mathcal{T}$ by parent lineage: $\{C_1, \ldots, C_m\}$
4: Sample cluster $C_j$ with weight $\max\left(\max_{e \in C_j} \mathrm{AUC}(e) - 0.5,\ 0\right)$
5: Sample experiment $e^* \in C_j$ with weight $\max(\mathrm{AUC}(e^*) - 0.5,\ 0)$
6: Retrieve related experiments via semantic nearest-neighbour and BM25 search
7: Retrieve ancestor chain of $e^*$
8: $d \leftarrow \textsc{ExploiterLLM}(e^*,\ \text{related experiments})$
9: **return** $d$

---

### A.2.2 Exploiter Agent Prompt Templates

---

**Exploiter Agent Prompt**

**System:**

You are an expert researcher improving Membership Inference Attack (MIA) signal designs. Deliver a targeted improvement to the current MIA signal design that is measurably more effective than the current and all prior attempts, while keeping computation and implementation lean. Prefer precision over verbosity; avoid full rewrites unless necessary. You should not propose a completely new MIA signal design, but rather improve the current design.

**User:**

Improve the current MIA signal design given the following function specifications and context:
`{experiment_context}`

**Function specifications:**
`{function_description}`

**Reference example** (format/style only—do NOT copy its approach):
`{example_python_code}`

**Relevant previous attempts:**
`{relevant_mia_signal_designs}`

**Current candidate:**
`{current_design}`

Return concise outputs:
- Failure modes (max 300 words, focus on why previous attempts failed to improve the MIA signal)
- High-level idea ($\leq$60 words)
- Design justification (max 300 words, focus on why the changes help and address previous failures)
- Implementation instructions

**Hard constraints** for the eventual code implementation:
- Python only, executable.

---

- Do NOT include any main/test functions in the code implementation.
- Easy to read and easy to modify.
- No extra model calls beyond inputs provided; avoid heavyweight resources.
- Strictly follow the function specifications and input/output specifications. Do NOT deviate from the function specifications and input/output specifications.

**Additional guidance:**
- The example code represents the current state of the art in MIA signal design. You MUST propose a better MIA signal design to outperform the SOTA.
- Please refer to the example code for format and input/output specifications.
- Make reasonable choices of hyperparameters.
- You should learn and reason from the previous attempts and the current design to make the new design more effective.
- Avoid trivial rewrites or generic confidence heuristics; highlight concrete algorithmic improvements.
- Keep instructions brief and focused on the core algorithm.

Output a JSON with exactly these fields:
1. `failure_modes`: Why previous attempts failed (max 300 words)
2. `idea`: The high-level idea of the MIA signal (max 100 words)
3. `design_justification`: The design justification (max 300 words, focus on why the changes help and address previous failures)
4. `implementation_instruction`: The implementation instruction of the MIA signal

## A.3 Code Agent

### Code Generation Agent Prompt

**System:**
You are a senior software engineer specializing in Python. Your job is to implement a Python function that computes a membership inference (MIA) signal score for a prompt-completion setting. Write clean, readable, and easily modifiable code that is fully executable.

**User:**
Implement the MIA signal function according to the spec and instructions below.

**Function spec:**
{function_description}

**Reference example** (for format/style only; do NOT copy its approach):
{example_python_code}

**High-level idea of the MIA signal:**
{idea}

**Implementation requirements:**
{implementation_instruction}

**Code requirements:**
- Return ONLY valid Python code block as plain text (no Markdown fences, no extra prose).
- Put all required imports at the top.
- Provide concise, high-level comments only where helpful. Avoid excessive or line-by-line comments.
- Implement the function exactly as specified (name/signature/return type).
- Do NOT include any main/test functions in the code implementation.
- Do NOT perform I/O (no printing, files, network) and do NOT rely on global state.
- Never return `None` or an empty response; always return a finite float for any input.
- Do NOT use try-except blocks to handle errors and exceptions.

- The environment is fixed. If the code block imports a library that is not installed, modify the code to not use that library.
- Think carefully about efficiency. If a pretrained model is used, consider declaring it as a global variable to avoid re-loading it multiple times.

**Output requirements:**
A single Python code block with the required function and all required imports at the top.

---

## Code Fix Agent Prompt

**System:**
You are a senior software engineer specializing in Python. Your job is to fix the code block that is provided to you. Write clean, readable, and easily modifiable code that is fully executable.

**User:**
Final goal: Implement a *new* MIA signal function according to the spec below.

**Function spec:**
{function_description}

**Reference example** (for format/style only; do NOT copy its approach):
{example_python_code}

**High-level idea of the MIA signal:**
{idea}

**Implementation requirements:**
{implementation_instruction}

**Current buggy code block:**
{code_block}

**Error message:**
{error_message}

**Guidance:**
- Given the error message, fix the code block to be executable and correct. Follow the original idea and implementation instructions as much as possible.
- If timeout, consider changing hyperparameters to reduce computation time.

**Code requirements:**
- Strictly follow the function specifications and input/output specifications. Do NOT deviate from them.
- Put all required imports at the top.
- Provide concise, high-level comments only where helpful. Avoid excessive or line-by-line comments.
- Implement the function exactly as specified (name/signature/return type).
- Implement ONLY the required function (no main/test functions).
- Do NOT perform I/O (no printing, files, network) and do NOT rely on global state.
- Never return `None` or an empty response; always return a finite float for any input.
- Do NOT use try-except blocks to handle errors and exceptions.
- The environment is fixed. If the code block imports a library that is not installed, modify the code to not use that library.
- Think carefully about efficiency. If a pretrained model is used, consider declaring it as a global variable to avoid re-loading it multiple times.

Output a JSON with exactly these fields:
1. `error_diagnosis`: A clear error summary
2. `changes_made`: The changes made to the code block

3. `code_block`: The complete fixed Python code block with the required function and all required imports at the top

## A.4 Executor Agent

The Executor Agent is responsible for executing the code generated by the Code Agent and returning the results. It uses a secure sandbox environment to run the code, captures any output or errors that occur during execution, and timeouts if the code takes too long to run. If successful, it returns the output; if an error occurs, it returns the last 20 lines of the standard error output to help the Code Agent debug and refine the code in subsequent iterations.

## A.5 Result Analyzer Agent

---

**Result Analyzer Agent Prompt**

**System:**
You are an expert researcher specializing in membership inference attacks (MIA) and machine learning security. Your role is to critically analyze MIA experiment results and provide comprehensive insights that will inform future research directions.

**User:**
Analyze the following MIA experiment and provide a structured summary of the findings.

**MIA Design Information:**
- Design Idea: `{idea}`
- Design Justification: `{design_justification}`
- Implementation Code: `{code_block}`
- Experiment Results: `{results}`
`{experiment_context}`

Your analysis should be concise and focus on the key points (max 300 words):
1. Evaluate the effectiveness of the MIA signal
2. Identify key insights about what makes this signal work or fail
3. Highlight limitations and potential failure modes
4. Note any novel or innovative aspects of the approach

---

## A.6 Experiment Harness

---
**MIA Signal Computation Function Specifications**

```
compute_mia_signal(generated_sample: Dict[str, Any]) -> float
- Compute a membership-inference signal for each text sample.
Inputs:
- generated_sample is a dict with:
Format:
generated_sample = {
    "original_text": "the prefix text ... the continuation text...",
    "prefix": "the prefix text ...",
    "ground_truth_suffix": "the ground-truth suffix text...",
    "suffix_generations": ["generated suffix 1", "generated suffix 2", ...]
}
`original_text`: The full target text, tokenized as whitespace-separated "words".
`prefix`: The prefix text (70% of the original text).
`ground_truth_suffix`: The ground-truth continuation (30% of the original text).
`suffix_generations`: A list of 100 model-generated continuations (strings)
↪   produced by sampling from the model given the same prompt prefix/context.

Output:
- A float number `mia_signal` that indicates the likelihood of membership. Higher
↪   signal values indicate higher likelihood of membership.

Requirements:
- The implementation must be self-contained and rely solely on the input parameters
↪   and common Python libraries.
- There are NO logits available. Only the above fields described in the format of
↪   `generated_sample` are available.
```
---

## A.7 Exploration-Exploitation Main Loop

---

**Algorithm 3** Dual-Agent MIA Main Loop

---

**Require:** budget $B$, config $C$, database $DB$

    **State:** $S = (\mathcal{D}, \mathcal{C}, \mathcal{R}, \mathcal{L})$ where

      $\mathcal{D} = (idea,\ rationale,\ instructions)$                        ▷ design

      $\mathcal{C} = (program,\ fix\_round)$                              ▷ code

      $\mathcal{R} = (status \in \{\texttt{ok}, \texttt{fail}, \texttt{timeout}\},\ error,\ metrics,\ analysis)$      ▷ run

      $\mathcal{L} = (iter,\ mode,\ parent\_id)$                            ▷ lineage

    ▷ *Seed iteration*

  1: $S \leftarrow \textsc{InitState}()$

  2: $\mathcal{D} \leftarrow (C.baseline\_idea,\ C.baseline\_rationale,\ \perp)$

  3: $\mathcal{C}.program \leftarrow C.baseline\_code$

  4: $\mathcal{L} \leftarrow (0,\ \texttt{seed},\ -1)$

  5: $\mathcal{R} \leftarrow \textsc{Execute}(\mathcal{C}.program)$

  6: $\mathcal{R}.analysis \leftarrow \textsc{Analyze}(S)$

  7: $\textsc{Insert}(DB, S)$

    ▷ *Search loop*

  8: **while** $\textsc{Count}(DB) < B$ **do**

  9:      $S \leftarrow \textsc{InitState}();\quad \mathcal{L}.iter \leftarrow \textsc{Count}(DB)$

      ▷ *Phase 1: Design*

10:      **if** $\mathcal{L}.iter \bmod 3 = 0$ **then**             ▷ Exploring new approaches every 3 iterations

11:          $(\mathcal{D}, \mathcal{L}) \leftarrow \textsc{Explore}(DB, C)$                    ▷ *parent* $=-1$

12:      **else**                               ▷ Improving existing ideas in other iterations

13:          $(\mathcal{D}, \mathcal{L}) \leftarrow \textsc{Exploit}(DB, C)$           ▷ *parent* $= retrieved\_experiment\_id$

14:      **end if**

      ▷ *Phase 2: Implement & validate*

15:      $\mathcal{C}.program \leftarrow \textsc{CodeGen}(\mathcal{D}, C)$

16:      $\mathcal{R} \leftarrow \textsc{Execute}(\mathcal{C}.program)$

17:      **while** $\mathcal{R}.status \neq \texttt{ok}$ **and** $\mathcal{C}.fix\_round < 3$ **do**

18:          $\mathcal{C}.program \leftarrow \textsc{CodeFix}(\mathcal{D},\ \mathcal{C}.program,\ \mathcal{R}.error)$

19:          $\mathcal{C}.fix\_round \leftarrow \mathcal{C}.fix\_round + 1$

20:          **if** $\mathcal{R}.status = \texttt{timeout}$ **then**

21:              $\mathcal{C}.fix\_round \leftarrow \mathcal{C}.fix\_round + 1$               ▷ timeout costs an extra retry

22:          **end if**

23:          $\mathcal{R} \leftarrow \textsc{Execute}(\mathcal{C}.program)$

24:      **end while**

25:      **if** $\mathcal{R}.status = \texttt{ok}$ **then**

26:          $\mathcal{R}.analysis \leftarrow \textsc{Analyze}(S)$

27:          $\textsc{Insert}(DB, S)$

28:      **end if**

29: **end while**

---

# B  Experiments and Results

## B.1  General Experiment Setup

For all experiment, we split the MIA dataset into 50% for training (used to design – running `AutoMIA`) and 50% for testing (used for final evaluation of the discovered signals). This make sure the discovered signals are not overfitted to the dataset during the searching stage. All the baselines and `AutoMIA` are evaluated on the same test set for a fair comparison. We run `AutoMIA` and OpenEvolve for 100 iterations with the same underlying LLM. The exploration and exploitation ratio is 1:2. Regarding the sandbox of the Executor Agent to run the generated code, we pre-installed common Python libraries such as NumPy, SciPy, and scikit-learn, torch, transformers, and some text processing libraries like NLTK and SpaCy. We acknowledge that some generated code may require additional libraries, but the current setup does not allow for dynamic installation of new packages for security and stability reasons.

## B.2  MIAs on black-box LLMs

### B.2.1  General Pipeline

The general pipeline for membership inference attacks (MIAs) on black-box large language models (LLMs) involves the following steps. The inference step is reused from the SOTA method (Hallinan et al., 2025).

Given a target model $M_\theta$, a test text $x$, a threshold $\epsilon$, a token index $k$, a number of samples $d$, and a MIA signal computation function $f$:

1. **Inference.** Use the prefix $x_{\leq k}$ as the prompt and sample $d$ outputs:

$$o^{(i)} \overset{\text{i.i.d.}}{\sim} M_\theta(\cdot \mid x_{\leq k}), \qquad i = 1, \ldots, d.$$

2. **Signal computation.** Compute the MIA signal using the generations $o_\theta^{(i)}$ and the ground-truth suffix $x_{>k}$:

$$S_\theta(x) := f\left(o_\theta^{(1)}, o_\theta^{(2)}, \ldots, x_{>k}\right).$$

3. **Decision.** Predict MEMBER if $S_\theta(x) > \epsilon$, otherwise predict NON-MEMBER.

### B.2.2  Human Baseline – Max Coverage Signal (Hallinan et al., 2025)

The SOTA method (Hallinan et al., 2025) uses the max coverage signal – the best performing signal among several designs proposed in their paper. The intuition is that if the target text $x$ is a member of the training data, then the model is more likely to generate completions that have high n-gram coverage with the ground-truth suffix $x_{>k}$.

Given $d$ sampled completions $\{o_\theta^{(i)}\}_{i=1}^d$ from $M_\theta(\cdot \mid x_{\leq k})$, the max-coverage signal is defined as follows.

$$f := \max_{i=1,\ldots,d} f^{(i)}, \qquad f^{(i)} := \text{Cov}_L\left(o_\theta^{(i)}, \ x_{>k}\right),$$

where $\text{Cov}_L(x_1, x_2)$ is the n-gram coverage score between two texts $x_1$ and $x_2$ at level $L$. For each token in $x_2$, we check the $L$-gram that ends at this token, and if it appears in $x_1$, we count it as a hit. The coverage score is the total number of hits divided by the total number of tokens in $x_2$.

### B.2.3  Geometric Edit-Distance Signal (by `AutoMIA`)

> **Raw High-level Idea by `AutoMIA`**
>
> Measure the geometric mean of two normalized scores: (1) median normalized Levenshtein distance between generations and ground truth (alignment), and (2) median normalized pairwise Levenshtein similarity among generations (consistency). Only high values in both indicate true memorization.

The best-performing signal discovered by `AutoMIA` for the Pythia 1.4B model on the ArXiv domain combines two scores via their geometric mean: proximity of the generations to the ground-truth suffix, and inter-generation consistency. Both scores are based on token-level edit distance.

**Token-level edit distance.** Let $\mathrm{ED}(a, b)$ be the Levenshtein edit distance between two sequences $a$ and $b$, capped at a maximum value $D_{\max} = 10$ for efficiency. We compute edit distance with the usual dynamic-programming recurrence (insertions, deletions, substitutions), clamping every cell to $D_{\max} + 1$ and terminating early if the minimum value of the current row exceeds $D_{\max}$.

The normalized edit distance is defined as:

$$\hat{d}(a, b) \;=\; \frac{\mathrm{ED}(a, b)}{\max(|a|, |b|)}$$

**Score computation.** Given $d$ sampled completions $\{o_\theta^{(i)}\}_{i=1}^d$ from $M_\theta(\cdot \mid x_{\leq k})$ and the ground-truth suffix $x_{>k}$, let $g_i$ and $r$ denote the token sequences obtained by whitespace-splitting $o_\theta^{(i)}$ and $x_{>k}$, respectively.

1. **Ground-truth proximity score.** Compute the normalized edit distances from each generation to the ground truth:
   $$\delta_i^{\mathrm{gt}} \;=\; \hat{d}(g_i, r), \qquad i = 1, \ldots, d.$$
   The first score is
   $$S_1 \;=\; 1 \;-\; \mathrm{median}\big(\delta_1^{\mathrm{gt}}, \ldots, \delta_d^{\mathrm{gt}}\big).$$

2. **Inter-generation consistency score.** Compute the pairwise normalized edit distances among all generations:
   $$\delta_{i,j}^{\mathrm{pw}} \;=\; \hat{d}(g_i, g_j), \qquad 1 \leq i < j \leq d.$$
   The second score is
   $$S_2 \;=\; 1 \;-\; \mathrm{median}\Big(\big\{\delta_{i,j}^{\mathrm{pw}}\big\}_{1 \leq i < j \leq d}\Big)$$

3. **MIA signal.** The final signal is the geometric mean of the two scores, clamped to $[0, 1]$:
   $$f \;=\; \mathrm{clamp}\Big(\sqrt{S_1 \cdot S_2}, \, 0, \, 1\Big).$$

Intuitively, $S_1$ is high when the model's generations closely resemble the ground-truth continuation (suggesting memorization), while $S_2$ is high when the generations are consistent with each other (suggesting the model has a concentrated predictive distribution over this prefix). The geometric mean requires both conditions to hold simultaneously for the signal to be large, which helps to reduce false positives that arise from either condition alone.

### B.2.4 Rare Trigram Aggregation Signal (by `AutoMIA`)

> **Raw High-level Idea by `AutoMIA`**
>
> Amplify membership signal by summing log(1 / (frequency_in_model * recurrence_count)) for trigrams appearing in ≥1 generation, where frequency_in_model is estimated from all generations across all samples – isolating trigrams that are both globally rare and locally reused in a single sample's generations.

The best-performing signal, discovered by `AutoMIA` for the Pythia 1.4B model on the Github dataset, aggregates inverse-frequency–weighted trigrams that appear across sampled generations.

**Signal computation.** Given $d$ sampled completions $\{o_\theta^{(i)}\}_{i=1}^d$ from $M_\theta(\cdot \mid x_{\leq k})$ and a precomputed global trigram frequency table $\text{freq}(\cdot)$ over a reference corpus, let $g_i$ denote the token sequence obtained by whitespace-splitting $o_\theta^{(i)}$.

1. **Trigram extraction.** For each generation $g_i$, extract the set of distinct trigrams $T_i = \{(g_i[t], g_i[t+1], g_i[t+2]) : t = 1, \ldots, |g_i| - 2\}$. Let $\mathcal{T} = \bigcup_{i=1}^d T_i$ be the union of all observed trigrams, and let the *recurrence count* of a trigram $\tau$ be the number of generations that contain it:

$$r(\tau) \;=\; |\{\, i : \tau \in T_i \}| \,.$$

2. **Weighted aggregation.** The MIA signal is the sum of log-inverse-frequency weights over all observed trigrams:

$$f \;=\; \sum_{\tau \in \mathcal{T}} \log \frac{1}{\text{freq}(\tau) \cdot r(\tau)},$$

where $\text{freq}(\tau)$ defaults to 1 for trigrams absent from the reference corpus.

If the signal is large, the generations contain rare trigrams that each appear in only a few of the $d$ samples. A memorized training example will cause the model to repeatedly produce unusual $n$-gram patterns that are globally infrequent, so amplifying the signal. For non-member texts, the generations tend to fall back on common, high-frequency trigrams that contribute little weight.

### B.2.5  Rarity-Weighted Longest-Match Signal (by `AutoMIA`)

> **Raw High-level Idea by `AutoMIA`**
>
> Compute the maximum normalized edit distance reduction between the ground-truth suffix and any generation, weighted by the rarity of the aligned subsequence in the ground truth – capturing partial memorization as low-cost correction of rare sequences.

This signal is discovered while running `AutoMIA` for Pythia 1.4B model on the Pubmed dataset. The signal considers the normalized edit distance and the longest contiguous match.

**Signal computation.** Given $d$ sampled completions $\{o_\theta^{(i)}\}_{i=1}^d$ from $M_\theta(\cdot \mid x_{\leq k})$ and the ground-truth suffix $x_{>k}$, let $g_i$ and $r$ denote the token sequences obtained by whitespace-splitting $o_\theta^{(i)}$ and $x_{>k}$, respectively.

1. **Ground-truth $n$-gram frequencies.** Collect all unigram, bigram, and trigram counts from $r$ into a single frequency table $c(\cdot)$, and let $N = \sum_\tau c(\tau)$ be the total count.

2. **Per-generation scoring.** For each generation $g_i$:

   (a) Compute the normalized Levenshtein distance:

   $$\hat{d}_i \;=\; \frac{\text{ED}(g_i, r)}{\max(|g_i|, |r|)}.$$

   (b) Find the *longest contiguous match*: the longest token span $\ell_i$ of length $\geq 2$ that appears as a contiguous block in both $g_i$ and $r$.

   (c) Compute a *rarity weight* based on the frequency of the matched span in the ground truth:

   $$w_i \;=\; \begin{cases} \dfrac{N}{c(\ell_i)} & \text{if } |\ell_i| \geq 2, \\[2mm] \dfrac{N}{|r|} & \text{otherwise (fallback)}. \end{cases}$$

(d) Combine into a per-generation score:

$$f^{(i)} \; = \; 1 \, - \, \hat{d}_i \cdot \left( 1 - \min\left( \frac{w_i}{N+1}, \; 1 \right) \right).$$

3. **MIA signal.** The final signal is the maximum over all generations:

$$f \; = \; \max_{i=1,\ldots,d} \; f^{(i)}.$$

The rarity weight $w_i$ is large when the longest contiguous match $\ell_i$ is an infrequent $n$-gram within the ground-truth suffix, indicating the model reproduced a distinctive rather than formulaic phrase. In this case the penalty factor $(1 - w_i/(N+1))$ shrinks toward zero, boosting $f^{(i)}$ toward 1. A generation that closely matches the ground truth (low $\hat{d}_i$) *and* reproduces a rare contiguous span thus receives the strongest membership signal. Taking the maximum over generations follows the same rationale as the max-coverage baseline: a single high-fidelity completion suffices as evidence of memorization.

### B.2.6 Inverse-Frequency Mismatch Signal (by `AutoMIA`)

> **Raw High-level Idea by `AutoMIA`**
>
> Measure the average inverse probability (1/p) of ground-truth tokens that are *missing or mismatched* in the closest 70% of generations, weighted by their rarity in the ground-truth suffix. Higher scores indicate memorization: the model consistently fails to reproduce rare tokens from the true suffix, revealing its rigid recall.

This signal (discovered for OPT7B, ArXiv) measures how well the model's generations reproduce the *rare* tokens of the ground-truth suffix. The key idea is that when a generation fails to match a token in the ground truth, the penalty is proportional to that token's inverse frequency within the suffix—so missing a rare, distinctive token costs more than missing a common one.

**Setup.** Let $r = (r_1, \ldots, r_L)$ be the token sequence of the ground-truth suffix, and let $p(t) = \#(t, r)/L$ be the empirical frequency of token $t$ in $r$. Define the inverse-frequency weight $w(t) = 1/p(t)$.

**Generation filtering.** For each of the $d$ sampled generations, compute the token-level Levenshtein distance to $r$, normalized by $L$. Sort the generations by this distance and retain the closest 70% (at least one), discarding outlier generations.

**Mismatch scoring.** For each retained generation $g = (g_1, \ldots, g_{L'})$, compute a position-wise mismatch score against the ground truth:

$$m(g) \; = \; \sum_{i=1}^{L} w(r_i) \cdot \mathbf{1}[\, i > L' \text{ or } g_i \neq r_i \,].$$

That is, each ground-truth position where the generation either has no token or has a different token incurs a penalty equal to the inverse frequency of the ground-truth token at that position.

**Final signal.** The MIA signal is the maximum mismatch score over all retained generations:

$$f \; = \; \max_{g \in \text{top-70\%}} \; m(g).$$

Higher values indicate that even the model's best generations fail to reproduce the suffix's rare tokens, which—perhaps counterintuitively—serves as the membership signal here: the score is largest when the ground truth contains many rare tokens that the model does not reproduce, and the threshold is calibrated accordingly.

### B.2.7 Recurrent Rare-Trigram Signal (by `AutoMIA`)

> **Raw High-level Idea by `AutoMIA`**
>
> Use the unnormalized sum of inverse-frequency weights for ground-truth trigrams that appear in at least two generations – amplifying rare, distinctive sequences reproduced consistently, without normalization that dilutes signal strength.

This signal asks: which ground-truth trigrams does the model *consistently* regenerate, and how rare are they? A trigram that is infrequent in the suffix yet appears across multiple independent generations is strong evidence of memorization.

**Setup.** Let $r = (r_1, \ldots, r_L)$ be the token sequence of the ground-truth suffix. Extract all trigrams $\mathcal{T} = \{(r_i, r_{i+1}, r_{i+2}) : i = 1, \ldots, L-2\}$ and let $c(\tau)$ be the number of times trigram $\tau$ occurs in $r$. Assign each unique trigram an inverse-frequency weight

$$w(\tau) = \frac{1}{1 + c(\tau)},$$

so that rarer trigrams receive higher weight.

**Recurrence counting.** For each of the $d$ sampled generations, extract its trigram set and check membership in $\mathcal{T}$. Let $a(\tau)$ be the number of generations that contain trigram $\tau$ at least once.

**Signal.** The MIA signal sums the inverse-frequency weights of all ground-truth trigrams that recur in at least two generations:

$$f = \sum_{\tau \in \mathcal{T}: \, a(\tau) \geq 2} w(\tau).$$

The threshold of two generations filters out coincidental single-generation matches, while the inverse-frequency weighting ensures that reproducing a distinctive phrase contributes more than reproducing a common one.

### B.2.8 Internal Repetition Signal (by `AutoMIA`)

> **Raw High-level Idea by `AutoMIA`**
>
> Measure the consistency of lexical repetition within each generation relative to the ground-truth suffix, using a normalized count of repeated n-grams (n=3-5) that appear at least twice within the same generation — capturing internal self-repetition as a signature of memorization.

This signal does not compare generations to the ground-truth suffix at all. Instead, it measures how repetitive each generation is *internally*: a model that has memorized a training example tends to produce outputs with repeated $n$-gram patterns, whereas generations for non-member prefixes are typically more varied.

**Per-generation repetition score.** For a generation $g = (g_1, \ldots, g_L)$, consider $n$-grams of sizes $n \in \{3, 4, 5\}$. For each $n$, let $c_n(\tau)$ denote the number of occurrences of $n$-gram $\tau$ in $g$. The raw repetition count is the total number of *excess* occurrences across all $n$-gram sizes:

$$R(g) = \sum_{n \in \{3,4,5\}} \sum_{\tau: \, c_n(\tau) \geq 2} (c_n(\tau) - 1),$$

normalized by the generation length to give $\hat{R}(g) = R(g)/L$.

**Signal.** The MIA signal is the average normalized repetition score across all $d$ generations:

$$f = \frac{1}{d} \sum_{i=1}^{d} \hat{R}(g_i).$$

### B.3   MIAs on Gray-box VLMs

#### B.3.1   Experiment Setup

We consider two settings: (1) image logits only and (2) caption logits only, to understand the privacy leakage of different modalities.

#### B.3.2   General Pipeline

The general pipeline for VLMs includes the following steps. The inference step is also reused from the SOTA method (Li et al., 2024).

Given a target VLM $M_\theta$, an image $x$, an associated caption $c$, a threshold $\epsilon$, and a MIA signal computation function $f$:

1. **Inference.** Use the image $x$ as the prompt and a fixed instruction `Describe this image` to generate a caption with logits:
$$o, \ell \;=\; M_\theta(\cdot \mid x \oplus \texttt{Describe this image}),$$
where $o = [o_1, o_2, ...]$ are the generated tokens for both image and text tokens, and $\ell = [\ell_1, \ell_2, ...]$ are the corresponding logits.

2. **Signal computation.** Compute the MIA signal using the generated caption $o$, the logits $\ell$, and the ground-truth caption $c$:
$$S_\theta(x) \;:=\; f(o, \ell, c)\,.$$

3. **Decision.** Predict MEMBER if $S_\theta(x) > \epsilon$, otherwise predict NON-MEMBER.

#### B.3.3   Human Baseline – Renyi Entropy Signal (Li et al., 2024)

Li et al. (2024) proposed the MaxRenyi by utilizing the Renyi entropy of the next-token probability on each image or text token. The intuition is that if the sample is a member of the training data, the model is more confident in generating the next token, leading to lower Renyi entropy.

**Renyi Entropy**   . The Renyi entropy of order $\alpha$ for a discrete probability distribution $P$ is defined as:

$$H_\alpha(P) = \frac{1}{1-\alpha} \log\left(\sum_i P(i)^\alpha\right),$$

where $\alpha > 0$ and $\alpha \neq 1$. As $\alpha \to 1$, the Renyi entropy converges to the Shannon entropy.

**MaxRenyi Signal.**   We get the average Renyi entropy for top $K\%$ of the tokens.

$$f \;=\; \frac{1}{|T|} \sum_{t \in T} H_\alpha(P_t),$$

where $T$ is the set of tokens corresponding to the top $K\%$ lowest Renyi entropy values among all generated tokens.

In pratice, we set $\alpha = 0.5$ and $K = 10\%$, which generally yields the best performance according to the original paper's findings.

#### B.3.4   Rank-Stability Signal (by `AutoMIA`)

> **Raw High-level Idea by `AutoMIA`**
>
> Memorization is signaled by the consistency of top-k token probability rankings across multiple forward passes with stochastic dropout, where member samples exhibit stable rank-orderings due to memorized deterministic patterns, while non-members show high rank variance from generalization noise.

The intuition is that for memorized inputs the model's top-token rankings are *stable* under small perturbations, whereas for non-member inputs the rankings are more sensitive to noise. This signal is found when running `AutoMIA` for the DALL-E dataset on the image-logit setting.

**Stochastic perturbation.** Given the logit tensor $\mathbf{z} \in \mathbb{R}^{L \times V}$ (sequence length $L$, vocabulary size $V$), perform $P = 5$ perturbed forward passes. In each pass $p$, add independent Gaussian noise to simulate dropout:

$$\tilde{\mathbf{z}}^{(p)} = \mathbf{z} + \boldsymbol{\epsilon}^{(p)}, \qquad \boldsymbol{\epsilon}^{(p)} \sim \mathcal{N}(0, 0.1^2\, \mathbf{I}).$$

Convert to probabilities via softmax and extract the indices of the top-$k$ tokens (with $k = 10$) at each sequence position. Concatenate these across positions into a single rank vector $\mathbf{r}^{(p)}$.

**Pairwise rank agreement.** For each pair of passes $(p, q)$, measure the disagreement between $\mathbf{r}^{(p)}$ and $\mathbf{r}^{(q)}$ via a normalized inversion count (Kendall-$\tau$ style):

$$d_{p,q} = \frac{\#\ \text{inversions between } \mathbf{r}^{(p)} \text{ and } \mathbf{r}^{(q)}}{\binom{k}{2}}.$$

**Signal.** The MIA signal is the negated mean pairwise distance:

$$f = -\frac{1}{\binom{P}{2}} \sum_{p<q} d_{p,q}.$$

Higher $f$ (i.e. lower rank disagreement) indicates the model's predictions are confident and stable under perturbation, suggesting the input was memorized.

### B.3.5 Positionally-Decayed Log-Ratio Variance Signal (by `AutoMIA`)

> **Raw High-level Idea by `AutoMIA`**
>
> Compute the MIA signal as the mean of the top 5% of position-decayed \*log-ratio gaps\* between the true token and its top-5 alternatives, where each gap is multiplied by the true token's own log-probability—amplifying only the most confident and consistent dominance events in early sequence positions.

This signal captures positions where the model's probability mass is unevenly distributed among the top alternatives relative to the true token, with an exponential bias toward earlier positions in the suffix.

**Log-ratio gaps.** Let $\mathbf{z}_i \in \mathbb{R}^V$ be the logit vector at position $i$ and let $t_i$ be the true token. Compute the log-probability of the true token under the full distribution, $\ell_i = \log p(t_i \mid \mathbf{z}_i)$. Then identify the top-5 alternative tokens (excluding $t_i$) by logit magnitude, and let $\tilde{\ell}_i^{(1)}, \ldots, \tilde{\ell}_i^{(5)}$ be their log-probabilities under a softmax restricted to just those five tokens. The log-ratio gap vector at position $i$ is

$$\mathbf{g}_i = \left(\ell_i - \tilde{\ell}_i^{(1)}, \ldots, \ell_i - \tilde{\ell}_i^{(5)}\right) \in \mathbb{R}^5.$$

**Positionally-decayed variance.** Compute the variance of each gap vector and apply an exponential position decay:

$$v_i = \mathrm{Var}(\mathbf{g}_i) \cdot e^{-i/8},$$

where $i$ is zero-indexed. The decay concentrates the signal on the first several tokens of the suffix, where memorization effects are strongest.

**Signal.** The MIA signal is the mean of the top 5% of the weighted variances $\{v_i\}_{i=0}^{L-1}$:

$$f = \text{mean}(\{v_i : v_i \geq Q_{95}(\{v_j\})\}),$$

where $Q_{95}$ denotes the 95th percentile. By focusing on the extreme tail, the signal isolates the few positions where the model's confidence structure is most anomalous—positions where the true token dominates some alternatives far more than others, suggesting it was seen during training.

### B.3.6 Top-$k$ Confidence Signal (by `AutoMIA`)

> **Raw High-level Idea by `AutoMIA`**
>
> The MIA signal is the mean of the top 10% of token-level top-5 log-probability *ranks*, not values. By ranking top-5 means across the sequence and selecting the highest-ranked tokens, we identify tokens where joint confidence is unusually high *relative to other tokens in the same sequence*, isolating true memorization clusters from fluent but non-memorized high-confidence regions.

This signal is a simple measure of how confidently the model concentrates probability mass on its top predictions.

**Per-position confidence.** Let $\mathbf{z}_i \in \mathbb{R}^V$ be the logit vector at position $i$. Compute the mean log-probability of the top-5 tokens under the full softmax:

$$\bar{\ell}_i = \frac{1}{5} \sum_{j=1}^{5} \log p\left(t_i^{(j)} \mid \mathbf{z}_i\right),$$

where $t_i^{(1)}, \ldots, t_i^{(5)}$ are the five highest-probability tokens at position $i$.

**Signal.** Select the top 10% of positions by $\bar{\ell}_i$ (i.e. the positions where the model is most confident) and return their mean:

$$f = \frac{1}{|\mathcal{I}|} \sum_{i \in \mathcal{I}} \bar{\ell}_i, \qquad \mathcal{I} = \left\{i : \bar{\ell}_i \geq Q_{90}(\{\bar{\ell}_j\})\right\}.$$

Higher values indicate that the model's most confident positions are *very* confident—its probability mass is sharply concentrated on a few tokens—which is characteristic of memorized inputs.

This signal is very simple and found to be effective for the Flickr image logits. It is worth noting that this signal is different from the Min-K% (Zhang et al., 2025), which calculates the log-probability of the ground-truth tokens.

### B.3.7 Neighbor-Entropy Contrast Signal (by `AutoMIA`)

> **Raw High-level Idea by `AutoMIA`**
>
> Use the models own logits to approximate token embeddings, but compute contrast only between the true token and its *top-k most similar neighbors in the same sequence*—not the full vocabulary. This isolates local discriminative suppression: members show high true logprob while suppressing nearby semantically coherent alternatives generated in the same context.

This signal contrasts the model's confidence on the true next token against the predictive uncertainty at nearby positions in a learned embedding space. The intuition is that for memorized text, the model assigns high probability to the true token *even when* contextually similar positions have high entropy, producing a large positive gap.

**Pseudo-embeddings and neighbor retrieval.** Let $\mathbf{z}_i \in \mathbb{R}^V$ be the logit vector at position $i$ (after the standard next-token shift). Define a pseudo-embedding $\mathbf{e}_i = \mathbf{z}_i[:128]/\|\mathbf{z}_i[:128]\|_2$ by $\ell_2$-normalizing the first 128 logit dimensions. Compute the cosine similarity matrix $\mathbf{S} = \mathbf{E}\mathbf{E}^\top$ (with self-similarities masked out) and let $\mathcal{N}_i$ be the set of $k = 5$ positions most similar to position $i$.

**Per-position contrast.** For each position $i$, compute:

1. The log-probability of the true next token: $\ell_i = \log p(t_i \mid \mathbf{z}_i)$.

2. The mean entropy across its neighbors: $H_i = \frac{1}{k} \sum_{j \in \mathcal{N}_i} H(\text{softmax}(\mathbf{z}_j))$, where $H(\cdot)$ is the Shannon entropy.

**Signal.** The MIA signal is the mean contrast across all positions:

$$f = \frac{1}{L} \sum_{i=1}^{L} (\ell_i - H_i).$$

A high value indicates that the model is confident on the true tokens ($\ell_i$ close to zero) while contextually similar positions carry high uncertainty ($H_i$ large)—a pattern characteristic of memorized sequences where the model has "locked in" specific continuations despite the context admitting many plausible alternatives. This signal is found for the Flickr caption logits, but it is not significantly better than the MaxRenyi baseline.

### B.4 Findings and Analyses

### B.4.1 MIA Diversity Analysis

To understand the diversity of the discovered MIAs, we first prompt the LLM to describe the MIA signal given the implementation code and then analyze embeddings of the descriptions, as detailed in Appendix. To avoid the bias of different description styles, we use the same prompt and force the generated description to be within the same format. We then use `Qwen3-Embedding-8B`, which is a leading embedding model, to encode the descriptions into vectors and analyze these embeddings. The prompt for generating the description is as follows:

---

**MIA Signal Description Generation Prompt**

Given the following Python code implementing a Membership Inference Attack (MIA) signal:

`{code}`

IGNORE completely: function signatures, imports, comments, docstrings, variable names, code style, helper functions, boilerplate, error handling, and the `compute_mia_signals` wrapper. Describe ONLY the executed algorithm inside `get_mia_signal`.

Describe the algorithm as a sequence of computational steps. Use exactly this 4-line format:

REPRESENTATION: What representation is extracted from the inputs? E.g., "token-level n-grams of order 3–5", "per-token log probabilities", "character-level edit operations", "embedding vectors"

COMPARISON: How are ground truth and generations compared? E.g., "set intersection over union (Jaccard)", "pointwise log-likelihood ratio", "cosine similarity of frequency vectors", "exact string match"

AGGREGATION: How are per-generation scores combined into one? E.g., "maximum across all generations", "geometric mean", "mean after inverse-frequency weighting", "any-match binary flag"

SCORE: What is the final score? E.g., "the aggregated similarity value directly", "log-ratio of target vs reference likelihood", "z-score relative to non-member distribution"

**Rules:**
- Two different implementations of the same algorithm MUST produce identical output.
- Two genuinely different algorithms MUST produce different output.
- Describe what the code DOES, not what comments SAY.

---

- Use plain lowercase. No hedging. No extra words beyond the 4 lines.
- Each line must be a single short phrase (under 20 words).

Here is an example of the MIA signal description generated by the LLM:

**Example Output**

```
representation:  token-level n-grams of order 3-5 from ground truth and each generation
comparison:  jaccard similarity between ground truth and each generation's n-grams, weighted by
inverse frequency of overlapping n-grams in the sample's generations
aggregation:  geometric mean of weighted jaccard scores across all generations
score:  geometric mean of weighted jaccard similarities
```

Fig. 7 presents the pairwise similarity within the set of MIAs discovered by each system (OpenEvolve and `AutoMIA`). The histogram shows that `AutoMIA`'s MIAs are clearly more diverse than OpenEvolve's MIAs, as `AutoMIA`'s distribution is left-skewed with more pairs having low cosine similarity. It is worth noting that the cosine similarity in general seems to be relatively high (mostly above 0.7), which may be due to the descriptions being generated in a similar style and within the same domain of MIA signals. However, the relative difference between the two distributions should be the main takeaway, which suggests that `AutoMIA` discovers more diverse MIAs than OpenEvolve.

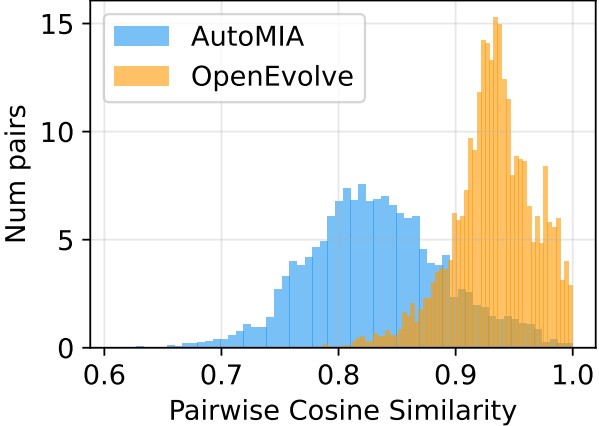

Figure 7: Histogram of cosine similarity the pairwise cosine similarity within the set discovered by OpenEvolve and `AutoMIA`. The less number of pairs with high cosine similarity, the more diverse the MIA set is. The histogram shows that `AutoMIA`'s MIAs are more diverse than OpenEvolve's MIAs, as `AutoMIA`'s distribution is left-skewed with more pairs having low cosine similarity.

Fig. 8 shows no clear correlation between the similarity and performance. Additionally, the PCA visualization of the embeddings of MIA signals discovered by `AutoMIA` is illustrated in Fig. 4b. Each point represents a MIA design, and the color indicates its performance (AUC). It does not show any clustering patterns, and each high-performing MIA is surrounded by low-performing MIAs in the PCA space. This suggests the complex landscape of MIA designs, where small changes in the design can lead to significant differences in performance. There is no single approach that performs well across all cases, and the performance of a MIA design can be sensitive to specific implementation details. Additionally, This highlights the importance of exploring a wide range of MIA designs and refining them based on empirical performance, as `AutoMIA` does, to discover effective signals that may not be intuitively obvious or closely related to existing methods.

### B.4.2  `AutoMIA` **with Target Context**

It is worth noting that in the main evaluation experiments, we do not provide the target context about the dataset or the target model. `AutoMIA` should be able to leverage its experiment attempts over time

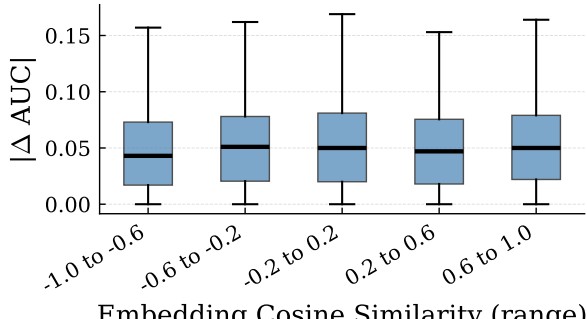

Figure 8: Cosine similarity between MIAs and their performance gap. Similarity does not predict performance.

to approach the right directions. However, if the context is provided, `AutoMIA` can directly focus on the right directions and avoid unnecessary attempts. In this experiment, we consider the Github dataset, which is fairly different from the natural language. Therefore, we expect that the target context can have more impact on this dataset than the human-language text datasets.

Without the context, the generated MIAs can be very general. The following is an example, where the MIA signal is based on an external large general corpus. More specifically, the LLM Agent decided to use Project Gutenberg (Gerlach & Font-Clos, 2018), which is a general corpus of English books.

> **Example High-level Idea without**
>
> Count total occurrences of each ground-truth n-gram (n=1,2,3) across all 100 generations, but weight each occurrence by the inverse frequency of that n-gram in a **large general corpus**—amplifying rare, distinctive fragments that signal true memorization over common phrases.

We provide the following context for the Explorer, Exploiter, and Analyzer when running `AutoMIA` on the Github dataset:

> **Context Provided to `AutoMIA`**
>
> Context of this experiment. Please account for this context to inform your design.
> - The LLM is Pythia 1.4B (deduped).
> - The dataset to be attacked is GitHub (code snippets) with average length of 200–300 tokens.

With the context, the LLM Agents actually leverage the information for their reasoning. For example, the following text is in the Analyzer's output: "... fails to overcome fundamental challenges: (1) **Code's syntactic regularity** causes high baseline prefix alignment even for non-memorized samples, diluting the signal; (2) **Tokenization via whitespace splitting is too coarse for code**, where indentation, variable names, and structure matter more than exact token sequences".

Here is another example from the Exploiter: "... **Extending n-grams to 10 tokens captures full code blocks (functions, loops)** ..."

### B.4.3 `AutoMIA` vs. Supervised MIAs

***Transferability across threat models.*** We train the supervised MIA on the source model (Pythia 1.4B) and evaluate it on both the source and a different target model (OPT 7B) on the Github dataset (Tab. 5). As expected, the supervised MIA transfers poorly across models. More notably, even in the ideal in-distribution

case—trained and evaluated on the same model and dataset—it does not outperform the human baseline, which uses the same n-gram feature set but a well-designed aggregation strategy.

| Method | Model | AUC | TPR@5%FPR |
|---|---|---|---|
| Human baseline | Source (Pythia 1.4B) | 0.664 | 0.209 |
| Supervised MIA | Source (Pythia 1.4B) | 0.630 | 0.134 |
| AutoMIA | Source (Pythia 1.4B) | **0.750** | **0.351** |
| Human baseline | Target (OPT 7B) | 0.620 | 0.157 |
| Supervised MIA | Target (OPT 7B) | 0.578 | 0.127 |
| AutoMIA | Target (OPT 7B) | **0.693** | **0.299** |

Table 5: Transferability across threat models. The supervised MIA is trained on the source model. It transfers poorly to the target model and, even on the source model, does not surpass the human baseline that shares its feature set.

***Transferability across benchmarks.*** We design signals on the MIMIR benchmark and evaluate them on WikiMIA-24 (Fu et al., 2024a) (Tab. 6). The supervised MIA does not generalize across benchmarks, whereas the unsupervised signals discovered by AutoMIA transfer substantially better.

| Method | Benchmark | AUC | TPR@5%FPR |
|---|---|---|---|
| Human baseline | WikiMIA-24 (len 64) | 0.557 | 0.085 |
| Supervised MIA | WikiMIA-24 (len 64) | **0.589** | 0.048 |
| AutoMIA | WikiMIA-24 (len 64) | 0.574 | **0.111** |
| Human baseline | WikiMIA-24 (len 128) | 0.525 | 0.061 |
| Supervised MIA | WikiMIA-24 (len 128) | 0.537 | 0.043 |
| AutoMIA | WikiMIA-24 (len 128) | **0.620** | **0.203** |
| Human baseline | WikiMIA-24 (len 256) | 0.630 | **0.114** |
| Supervised MIA | WikiMIA-24 (len 256) | 0.565 | 0.041 |
| AutoMIA | WikiMIA-24 (len 256) | **0.666** | **0.114** |

Table 6: Transferability across benchmarks. Signals are designed on MIMIR and evaluated on WikiMIA-24. The unsupervised nature of AutoMIA generalizes better across benchmarks than the supervised MIA.

### B.4.4 Exploration-Exploitation Ratio Analysis

We conduct an experiment on the black-box LLM MIA setting. We vary the exploration-exploitation ratio in the AutoMIA framework. Each ratio produces a set of 100 proposed MIA designs. We then analyze the performance of sets at different percentiles (median, 90th, and top-performing) of the proposed MIAs. The results are shown in Fig. 9. We find that an exploration-exploitation budget allocation of 1:2 (one-third exploration and two-thirds exploitation) consistently yields the best performance across different percentiles of the proposed MIAs, for both median and top-performing attacks. This suggests that a balanced approach that allows for sufficient exploration while still leveraging exploitation of promising designs is effective in discovering high-performing MIA signals.

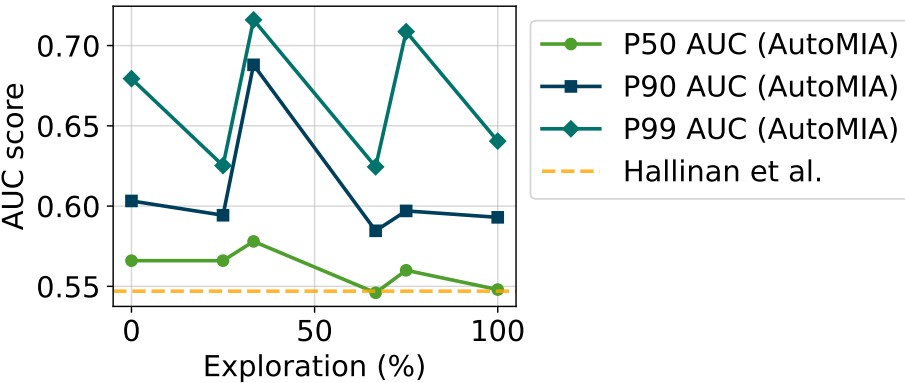

Figure 9: Exploration-Exploitation Ratio Analysis. An exploration-exploitation budget allocation of 1:2 (one-third exploration and two-thirds exploitation) consistently yields the best performance across different percentiles of the proposed MIAs, for both median and top-performing attacks.

