# OpenReview forum: "Automated Membership Inference Attacks: Discovering MIA Signal Computations using LLM Agents"
_TMLR — Under review for TMLR_

### Review · Reviewer_VYnJ · 2026-05-17

**Summary Of Contributions:**

This paper presents AutoMIA, an agentic system that automatically designs and implements membership inference attack (MIA) signal computation methods, achieving higher attack performance than existing human-designed and algorithmic baselines on both black-box large language models and gray-box vision-language models. Specifically, the authors first introduce a design phase with two types of LLM agents: the Explorer, which generates novel MIA signal designs through a novelty-guided iterative loop, and the Exploiter, which refines high-performing designs from past experiments using performance-guided reasoning. Second, the Programmer Agent translates the proposed designs into executable Python code, while the Executor Agent runs the code and collects performance metrics, forming a complete evaluation pipeline. Third, the Analyzer Agent evaluates the experimental results and stores them in a shared database, which the design agents later retrieve to guide further exploration and exploitation. Finally, through an iterative evolutionary loop with a 1:2 exploration-to-exploitation ratio, AutoMIA progressively discovers more effective MIA signal functions, as demonstrated by consistent AUC improvements over baselines and diverse, transferable MIA designs across datasets. However, I have some concerns about this paper. My detailed comments are as follows.


*Strengths:*

1. The authors develop an agentic framework, AutoMIA, that automates the design and implementation of membership inference attack (MIA) signal computation methods using multiple cooperating LLM agents.
2. AutoMIA’s design emphasizes natural language-level reasoning before code generation, enabling more flexible and diverse MIA signal discovery compared with direct code-level evolutionary approaches.
3. The experimental evaluation is comprehensive, covering both black-box large language models and gray-box vision-language models, with ablation studies and PCA analyses that demonstrate the effectiveness, diversity, and transferability of the discovered MIA signals.


*Weaknesses:*

1. Although the paper claims AutoMIA is designed for MIA signal computation, it does not explain how the agent framework is tailored to MIA-specific characteristics. The authors should clarify how Explorer and Exploiter leverage properties unique to MIA.
2. The fixed 1:2 Explorer-to-Exploiter ratio is hard-coded without sensitivity analysis or theoretical justification, leaving the evidence for the efficiency and effectiveness of the evolutionary search incomplete.
3. While the paper reports metrics such as AUC, it lacks statistical significance testing and detailed error analysis, which is especially important given the inherent randomness in model outputs.
4. Experiments are conducted on a small set of text and image datasets; generalization to other domains, larger models, or industry-relevant data is unclear.
5. The overall method description does not clearly emphasize design choices specific to MIA, and the framework appears general enough to potentially support other tasks such as automated text-generation detection[A][B][C]. The authors should discuss the broader applicability and potential extensions of their system beyond MIA.


*Reference:*

[A] Detecting Machine-Generated Texts by Multi-Population Aware Optimization for Maximum Mean Discrepancy. ICLR 2024.

[B] HLD: Approximate Hierarchical Linguistic Distribution Modeling for LLM-Generated Text Detection. ICLR 2026.

[C] Learning From Dictionary: Enhancing Robustness of Machine-Generated Text Detection in Zero-Shot Language via Adversarial Training. ICLR 2026.

**Audience:**

Yes

**Audience Explanation:**

AutoMIA demonstrates how LLM agents can be used to automate the design and optimization of membership inference attack signals, which is relevant for researchers in privacy auditing, machine learning security, and LLM evaluation. Even though the method focuses on signal computation and the experiments are limited to a few datasets and models, the approach introduces a general framework and insights that could inspire further work on automated attack discovery, robustness evaluation, or privacy analysis, making the results meaningful for a subset of TMLR’s audience.

**Claims And Evidence:**

Yes

**Claims Explanation:**

The submission provides partial evidence that AutoMIA can automatically discover high-performing MIA signal functions, as shown by AUC metrics and ablation studies. However, the evidence is limited because the framework’s MIA-specific design is unclear, the 1:2 Explorer-to-Exploiter ratio lacks justification, and there is no statistical significance testing or error analysis despite randomness in model outputs. Experiments cover only a small set of datasets, leaving generalization to other domains or larger models uncertain. Additionally, the framework appears general enough to support tasks beyond MIA, but broader applicability is not discussed. Overall, results are convincing within the tested scope, but more analysis and discussion are needed to fully support the claims.

**Requested Changes:**

See Weaknesses.

---

> ### Author Response · Authors · 2026-07-13
> **Response to Reviewer VYnJ**
>
> First of all, we would like to thank the reviewer for their time and effort in reviewing our paper. We appreciate your constructive comments and suggestions. Below we provide detailed responses to each of your concerns.
>
> ## Weaknesses 1 & 5
> We thank the reviewer for raising these points. While the key difference between AutoMIA and general-purpose code search (AlphaEvolve) is that AutoMIA reasons on the design space using natural language while AlphaEvolve reasons on the raw code space. We agree that the meta-architecture can be general; like most agentic search systems, our AutoMIA can be re-targeted to other domains through configuration and prompting. In the new version of the paper, we have removed "specialized systems" from the contribution statement and added a discussion regarding the potential extensions of AutoMIA.
>
>
> ## Weakness 2
> Thank you for the great suggestion, we have added an experiment in the new version of the paper (Figure 9) to analyze the effect of the exploration-exploitation ratio on the performance of AutoMIA. We vary the exploration-exploitation ratio from 0.0 to 1.0. For each ratio, we run AutoMIA to produce 100 MIAs. We then analyze the average and top 10/top 1 percentile performance of the MIAs. We find that the exploration-exploitation ratio has a significant impact on the performance of AutoMIA. The results show that spending 1/3 of the budget on exploration and 2/3 on exploitation yields the best performance in both average and top percentile performance.
>
> ## Weakness 3
> Due to the computational cost, we unfortunately cannot afford to repeat all experiments multiple times. However, we ensure that all the methods are evaluated on the same train/test split for a fair comparison. Here we run AutoMIA 5 times for the black-box LLM MIA for ArXiv datasets. While the human baseline AUC score is around 0.54, AutoMIA provides MIAs with AUC score: $ 0.721 \pm 0.011 $. The p-value of the t-test between AutoMIA and the human baseline is $3.7 \times 10^{-6}$, which indicates that AutoMIA robustly outperforms the human baseline for this setting.
>
>
> ## Weakness 4
> We appreciate this comment. In our experiments, we have used well-known benchmarks [1,2]. These benchmarks have been widely used in the MIA literature in the past two years. Additionally, we have also discussed the limitation of our benchmarks, leaving the industry-scale and computationally-expensive evaluation for future work.
>
> ---------
> [1] Duan, Michael et al. “Do Membership Inference Attacks Work on Large Language Models?” ArXiv abs/2402.07841 (2024). 193 citations
>
> [2] Li, Zhan et al. “Membership Inference Attacks against Large Vision-Language Models.” ArXiv abs/2411.02902 (2024). 54 citations

---

### Review · Reviewer_2g1J · 2026-05-18

**Summary Of Contributions:**

# Summary:

The authors factor the standard MIA pipeline into two stages: *inference* (querying the target model and collecting outputs) and *signal computation* (mapping those outputs to a scalar score that separates members from non-members). They fix the inference stage to match existing protocols and target only the signal-computation stage.

The authors propose AutoMIA is an evolutionary search loop in which LLM agents iteratively propose, implement, evaluate, and refine signal functions. A shared database stores every attempt as a tuple of (design, code, result, analysis), retrievable by dense embeddings, BM25, or tree lineage. Two classes of agents drive the loop.

The *Explorer* proposes novel designs by sampling random prior experiments, generating an initial candidate, and refining it through a critic loop with a Novelty Judge LLM that issues accept/revise/redesign verdicts. The *Exploiter* improves existing top-performers: it samples a parent design with probability proportional to |AUC - 0.5|, retrieves the parent's ancestor chain, siblings, and semantically related designs, and asks the LLM to first articulate failure modes of prior attempts before proposing a child.

 A *Programmer* compiles natural-language designs into Python; an *Executor* runs the code; an *Analyzer* writes a a summary back into the database, where it becomes prior art for future agents.

## Empirical results

AutoMIA is evaluated on two settings against the respective SOTA human-designed baselines and OpenEvolve (an AlphaEvolve reimplementation that evolves raw code). All methods share the same inference protocol; only the signal function differs. he largest gain is on OPT-7B/ArXiv, where AUC improves from 0.54 to 0.70 and TPR@1%FPR from 0.02 to 0.10. AutoMIA also consistently beats OpenEvolve.



# Some weakness:
- A central weakness is the uniform reliance on a labeled auxiliary set Dtrain of roughly **half** the benchmark, with no ablation of how performance degrades as it shrinks. Selecting the best of 100 candidates on a noisy AUC estimator inflates the reported gain, and the test split only partially absorbs this. The realistic attacker regime, i.e.,  small or no labeled probe set, is exactly where this bias would dominate, and it is untested. The headline gain therefore conflates genuine signal discovery with selection under a generous auxiliary-data assumption.


- If AutoMIA isn't discovering new *principles* — only new *combinations and parameterizations* within a known feature family (n-gram overlap, edit distance, rarity weighting, cross-generation consistency, top-k confidence) — then the same gains should be reachable by:

1. Enumerating a rich feature bank from the known principles: for each example, compute dozens of features (n-gram coverage at various L, normalized edit distance, pairwise generation similarity, trigram rarity weights, longest contiguous match, internal repetition rates, top-k log-prob aggregates, etc.).
2. Fitting a supervised classifier on D_train to combine them into a single membership score.
3. Reporting test AUC.

This is *exactly* the same problem AutoMIA solves: supervised learning of a scalar score from labeled (member, non-member) pairs. Logistic regression, gradient boosting, or a small MLP on a few dozen hand-designed features would almost certainly match or beat the AutoMIA winners, with vastly less compute and no LLM API calls.

If that's true, the LLM is doing feature selection by elimination over a small discrete set, which is something a forward-selection wrapper or L1-regularized regression handles.

- The evaluation is restricted to two recent and acknowledgedly under-developed MIA settings (MIMIR black-box LLM, gray-box VLM logits), where current human baselines achieve modest AUCs and no calibrated ceiling exists. A more diagnostic test would include at least one classification setting with shadow-model access, where LiRA-style attacks provide a near-optimal reference: matching, approaching, or exceeding LiRA would tell us whether AutoMIA discovers genuine signal or merely automates search over an under-explored design space.

Typos

Table 3 caption: EvoMIA--> AutoMIA
"AutoMIA boost the AUC of the image logits from"

**Audience:**

Yes

**Audience Explanation:**

Yes. The direction of using agents for refining hypothesis and suggesting new hypothesis is an active area of research, and, bringing it in privacy research can be of broad interest by the privacy community.

**Broader Impact Concerns:**

No.

**Claims And Evidence:**

Yes

**Claims Explanation:**

Yes the authors provide enough ablation studies and they are precise about various stage of the system.

**Requested Changes:**

The main changes are based on the weakness i discussed above. Each of the weakness may involve new experiments or discussion added to the paper.

---

> ### Author Response · Authors · 2026-07-12
> **Response to Reviewer 2g1J (1/2)**
>
> We thank the reviewer for their valuable comments and the time spent reviewing our manuscript. We would like to clarify the main motivation behind AutoMIA is just to replace the manual efforts in the current MIA design pipeline with LLM Agents.
>
> Therefore, regarding the train/test data, we would like to clarify that the same concern can also apply to human-designed MIAs. In the manual design process, researchers often explore multiple strategies, design choices, and hyperparameters before selecting the proposed MIA. If the final attack is evaluated on the same data used during this design process, the reported performance may be optimistic. Despite this, we acknowledge that the MIA community has not been doing this with a common assumption that unsupervised MIA is well generalized beyond the benchmark on which it was developed. In our setting, as a entirely new approach, to avoid optimistic performance estimation, we strictly perform the train/test split.
>
> To compare with supervised MIAs, the key difference is around transferability. The supervised MIA here use n-gram features, which is inspired by Hallinan et al. paper.
>
> **1. Transferability across threat models.**
>
> | Method | LLM | AUC | TPR@5%FPR |
> | --- | --- | --- | --- |
> | Human basedline | Source (Pythia 1.4B) | 0.664 | 0.209 |
> | Supervised MIA | Source (Pythia 1.4B) | 0.630 | 0.134 |
> | AutoMIA | Source (Pythia 1.4B) | 0.750 | 0.351 |
> | Human basedline | Target (OPT 7B) | 0.620 | 0.157 |
> | Supervised MIA | Target (OPT 7B) | 0.578 | 0.127 |
> | AutoMIA | Target (OPT 7B) | 0.693 | 0.299 |
>
> While it is intuitive that the supervised MIA is not transferable well, even for the ideal case training on the same LLM and the same dataset, the supervised MIA does not necessarily outperform the human baseline which employs the same feature set but with a well-designed aggregation strategy.
>
> **2. Transferability across benchmarks.** We perform AutoMIA and supervised MIA on the MIMIR benchmark then evaluate the performance on the WikiMIA-24 benchmark [1].
>
> | Method | Benchmark | AUC | TPR@5%FPR |
> | --- | --- | --- | --- |
> | Human basedline | WikiMIA-24 (length = 64) | 0.557 | 0.085 |
> | Supervised MIA | WikiMIA-24 (length = 64) | 0.589 | 0.048 |
> | AutoMIA | WikiMIA-24 (length = 64) | 0.574 | 0.111 |
> | Human basedline | WikiMIA-24 (length = 128) | 0.525 | 0.061 |
> | Supervised MIA | WikiMIA-24 (length = 128) | 0.537 | 0.043 |
> | AutoMIA | WikiMIA-24 (length = 128) | 0.620 | 0.203 |
> | Human basedline | WikiMIA-24 (length = 256) | 0.630 | 0.114 |
> | Supervised MIA | WikiMIA-24 (length = 256) | 0.565 | 0.041 |
> | AutoMIA | WikiMIA-24 (length = 256) | 0.666 | 0.114 |
>
> As the common assumption in the MIA community, we show that the unsupervised nature of AutoMIA allows it to generalize better across target models and benchmarks. Therefore, we do not consider supervised MIA as a direct baseline for AutoMIA.

---

> ### Author Response · Authors · 2026-07-12
> **Response to Reviewer 2g1J (2/2)**
>
> Additionally, we would like to emphasize that feature engineering and aggregation strategies are crucial contributions for the field of MIA. For example, in the LLM MIA literature, the Min-K% attack [2] proposed to consider only the top K% of tokens instead of all tokens as the standard attack using the PPL/loss value as MIA score. Min-K%++ [3] and DC-PDD [4] both inherit the idea of Min-K% but further added a calibration factor to the signal. Therefore, exploring different feature engineering and methods to aggregate the signal has been an important process to enhance MIA performance.
>
>
> -----
>
> [1] MIA-Tuner: Adapting Large Language Models as Pre-training Text Detector, Fu et al., AAAI 2025.
>
> [2] Min-K%: Detecting Pretraining Data from Large Language Models, Shi et al, ICLR 2024.
>
> [3] Min-K%++: Improved Baseline for Pre-Training Data Detection from Large Language Models, Zhang et al., ICLR 2025 (spotlight)
>
> [4] Pretraining Data Detection for Large Language Models: A Divergence-based Calibration Method, Zhang et al., EMNLP 2024

---

### Review · Reviewer_29oa · 2026-06-27

**Summary Of Contributions:**

This paper proposes AutoMIA, an LLM-agent-driven framework to discover new candidate membership inference attacks (MIAs) for LLMs. The framework itself is straightforward, allowing agents to iteratively run MIA experiments and reason about the results to improve the attack.

Strengths:
* The authors accurately identify that designing membership inference attacks often involves manually searching for signals that distinguish member and non-member data. It is worthwhile to try to automate this process using LLM agents.
* Empirically, the discovered attacks outperform human-designed attacks on both AUC and TPR at low FPR metrics. Qualitatively, these are not obvious attacks.

Weaknesses:
* My main concern is the choice of VLM datasets. Unfortunately, the VLM datasets used in this paper suffer the distribution-shift weaknesses pointed out by Duan et al. as well as Das et al. [1]. Unfortunately this issue was not adequately caught or addressed by the reviews of the prior work that proposed this dataset. I do not necessarily expect that the authors will propose an entirely new dataset that addresses these distribution-shift issues in the current work. But it is important to note this potential evaluation issue in the paper and qualify the claims regarding the performance on VLMs. The best resolution would be to run a blind baseline attack for these datasets (following the Das et al. work) and include that alongside the results, which would be valuable to the community and others who may wish to use these datasets and contextualize the current results.

[1] https://arxiv.org/pdf/2406.16201

**Audience:**

Yes

**Audience Explanation:**

Yes, MIAs for LLMs should be of interest to at least some of the TMLR audience.

**Claims And Evidence:**

No

**Claims Explanation:**

For LLMs, yes the claims are supported by convincing evidence; for VLMs, see weaknesses above.

**Requested Changes:**

* Please see Weaknesses above regarding the evaluation on VLMs. I would hesitate to recommend acceptance if this issue is not addressed.
* I would like to clarify the experimental setup. My understanding in the LLM setting is the following: AutoMIA takes as input the train set for one of the datasets, performs search, outputs *only* the top performing MIA for that dataset on the training set, and the performance reported in Table 1 is the performance of that MIA on the test set. This is repeated separately for each dataset. Is this correct? Or, does AutoMIA output multiple candidate MIAs and the reported performance is of the best performer on the test set?

---

> ### Author Response · Authors · 2026-07-20
> **Response to Reviewer 29oa**
>
> We would like to thank the reviewer for their time and effort in reviewing our paper.
>
> Regarding the concern about the VLM datasets, we have added blind-test results for the VLM benchmarks, along with a discussion in the revised paper highlighting the distribution shift between the member and non-member samples (Section 4.1 & Table 2).
>
>
> While these VLM benchmarks exhibit a significant distribution shift, we hope the reviewer will view AutoMIA as a general framework for automatically designing membership inference attacks, rather than one tailored to a specific benchmark. To further demonstrate its generality, we evaluate AutoMIA on a recent MIA benchmark for Large Reasoning Models (LRMs) introduced in January 2026 [1]. We consider two configurations of AutoMIA: (1) a dataset-specific version, where the evaluation objective is the AUC on a single dataset, and (2) a combined version, where the evaluation objective is the average AUC across both datasets. As shown below, AutoMIA consistently improves upon the recent state-of-the-art attack (BlackSpectrum) and can be readily adapted to optimize performance across multiple datasets through a simple change to its evaluation objective.
>
> | Dataset | Method | AUC | TPR@1% FPR | TPR@5% FPR |
> |---------|--------|:---:|:----------:|:----------:|
> | **ArXiv** | reasoning_len | 0.507 | 0.005 | 0.040 |
> |  | gzip_ratio | 0.553 | 0.016 | 0.069 |
> |  | neg_gpt2_nll | 0.585 | 0.024 | 0.079 |
> |  | blackspectrum | 0.777 | 0.077 | 0.311 |
> |  | AutoMIA | **0.827** | **0.129** | **0.404** |
> |  | AutoMIA (combined) | 0.786 | 0.061 | 0.332 |
> |  |  |  |  |  |
> | **Book** | reasoning_len | 0.542 | 0.000 | 0.054 |
> |  | gzip_ratio | 0.540 | 0.014 | 0.090 |
> |  | neg_gpt2_nll | 0.693 | 0.107 | 0.239 |
> |  | blackspectrum | 0.799 | 0.082 | 0.295 |
> |  | AutoMIA | **0.843** | **0.185** | **0.433** |
> |  | AutoMIA (combined) | 0.821 | 0.139 | 0.386 |
>
>
> Regarding the reviewer's question, we would also like to clarify that AutoMIA outputs many candidate MIAs rather than a single attack. For example, each point in Figure 4 represents one MIA proposed by AutoMIA. In Table 1 of the paper, we run AutoMIA independently on each dataset and report the best-performing attack discovered for that dataset. However, AutoMIA can be naturally extended to multi-dataset settings by modifying the evaluation objective, as demonstrated by the "AutoMIA (combined)" results above. Additionally, all results are evaluated on the held-out test sets.
>
> [1] https://arxiv.org/abs/2601.13607